 

# Binding stoichiometry and structural model of the HIV-1 Rev/importin *β* complex

Didier Spittler[1], Rose-Laure Indorato[1], Elisabetta Boeri Erba[1], Elise Delaforge[1], Luca Signor[1], Simon J Harris[1], Isabel Garcia-Saez[1], Andrés Palencia[2], Frank Gabel[1], Martin Blackledge[1], Marjolaine Noirclerc-Savoye[1], Carlo Petosa[1]

**HIV-1 Rev mediates the nuclear export of intron-containing viral RNA transcripts and is essential for viral replication. Rev is imported into the nucleus by the host protein importin *β* (Imp*β*), but how Rev associates with Imp*β* is poorly understood. Here, we report biochemical, mutational, and biophysical studies of the Imp*β*/Rev complex. We show that Imp*β* binds two Rev monomers through independent binding sites, in contrast to the 1:1 binding stoichiometry observed for most Imp*β* cargos. Peptide scanning data and charge-reversal mutations identify the N-terminal tip of Rev helix *α*2 within Rev's arginine-rich motif (ARM) as a primary Imp*β*-binding epitope. Cross-linking mass spectrometry and compensatory mutagenesis data combined with molecular docking simulations suggest a structural model in which one Rev monomer binds to the C-terminal half of Imp*β* with Rev helix *α*2 roughly parallel to the HEAT-repeat superhelical axis, whereas the other monomer binds to the N-terminal half. These findings shed light on the molecular basis of Rev recognition by Imp*β* and highlight an atypical binding behavior that distinguishes Rev from canonical cellular Imp*β* cargos.**

## Introduction

The HIV-1 protein Rev (regulator of expression of the virion) is an RNA-binding protein essential for the production of mature viral particles (1, 2) (reviewed in references 3, 4, and 5). The initial RNA molecule that results from proviral DNA transcription is differentially processed into multiply spliced transcripts that encode Rev and other early-stage viral proteins, as well as partly spliced and unspliced transcripts that encode viral structural and enzymatic proteins and provide genomic RNA for encapsidation. Host cell mechanisms rapidly export fully spliced transcripts to the cytoplasm but retain incompletely spliced molecules in the nucleus

(6, 7). Rev circumvents these mechanisms by mediating the nuclear export of intron-containing viral transcripts (8, 9, 10). Rev binds to these transcripts by recognizing the Rev-response element (RRE), a ~350-nucleotide region that adopts a complex stem-loop structure (8, 9, 11, 12). A first Rev monomer binds with high affinity to a small region of the RRE, stem loop IIB, triggering the cooperative binding of additional Rev monomers to the RRE (13, 14). The Rev/RRE complex is then exported to the cytoplasm by the nuclear export factor CRM1 (15, 16, 17), allowing for synthesis of late-stage viral proteins and packaging of the replicated viral genome into new viral particles. Other reported roles for Rev include regulating the splicing (18, 19, 20), translation (21, 22, 23), and encapsidation (24, 25, 26) of viral RNAs; modulating the stability or localization of the HIV-1 proteins Tat (27) and integrase (IN) (28, 29); and interacting with host cell molecules (30), such as RNA helicases and hnRNPs (31, 32, 33, 34, 35).

To bind RRE-containing viral transcripts and perform other nuclear functions, Rev must first be imported from the cytosol, a step mediated by host cell proteins of the karyopherin/importin-*β* family (reviewed in references 36, 37, 38, 39, 40, and 41). Different members of this family have been implicated as potential Rev import factors, including importin *β* (Imp*β*), transportin-1, transportin-5, and transportin-7 (42, 43, 44, 45). One study reported that Rev is imported by Imp*β* in T-lymphocyte–derived (293T, Jurkat and CEM) cell lines and by transportin-1 in HeLa cells and in a monocytic (THP-1) and histocytic (U937) cell line, revealing that the Rev import pathway is host cell–dependent (45). Given the central role of CD4[+] T lymphocytes for HIV-1 biology, the present study focuses on the Imp*β* pathway.

Imp*β* heterodimerizes with importin *α* (Imp*α*) to import proteins bearing a classical NLS, which typically contains one or two clusters of basic residues (37). Imp*α* uses its C-terminal domain to recognize the NLS (46, 47) and its N-terminal Imp*β*-binding (IBB) domain to associate with Imp*β* (48). HIV-1 Rev bypasses this pathway by directly interacting with Imp*β* and is imported independently of Imp*α* (42) (Fig 1A). Imp*β* is regulated by Ran, a Ras-related GTPase which is predominantly bound to GTP in the nucleus and to GDP in the cytosol. Imp*β* associates with its cargo (directly or via Imp*α*) in the

[1]Université Grenoble Alpes, Commissariat à l'Énergie Atomique et aux Énergies Alternatives (CEA), Centre National de la Recherche Scientifique (CNRS), Institut de Biologie Structurale, Grenoble, France   [2]Institute for Advanced Biosciences, Structural Biology of Novel Targets in Human Diseases, INSERM U1209, CNRS UMR5309, Université Grenoble Alpes, Grenoble, France

Correspondence: marjolaine.noirclerc@ibs.fr; carlo.petosa@ibs.fr

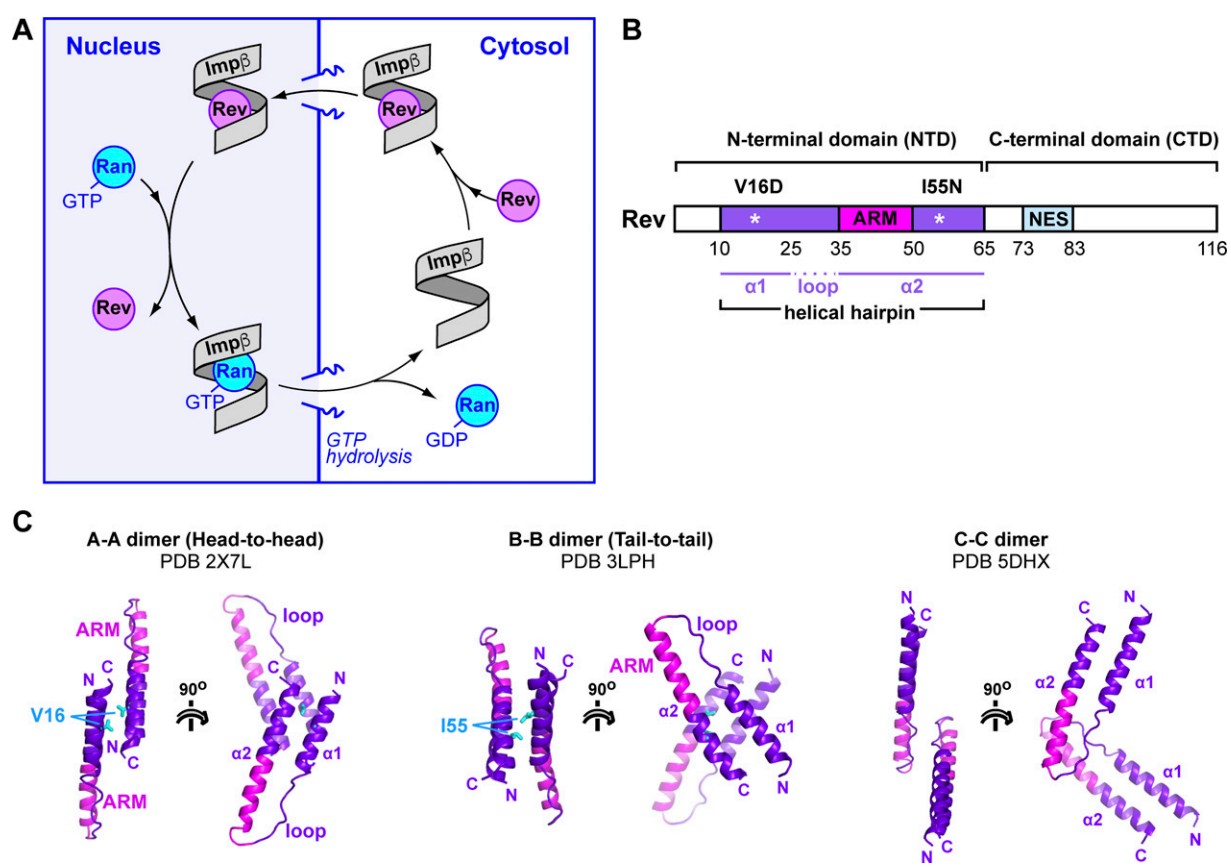

**Figure 1.  Rev structure and dimerization interfaces.**
**(A)** Scheme depicting the nuclear import cycle of Rev by Impβ. **(B)** Domain organization of Rev. **(C)** Oligomerization interfaces reported for Rev. Residues mutated to destabilize the A–A and B–B interfaces are indicated in cyan.

cytosol, translocates through the nuclear pore complex via favorable interactions with FG-repeat containing nucleoporins, and releases the cargo in the nucleus upon binding RanGTP. Atomic structures are known for Impβ in the unbound state (49, 50) and in complex with Ran (51, 52, 53), nucleoporins (54, 55, 56), importin7 (57), and diverse macromolecular cargos (48, 57, 58, 59, 60, 61, 62, 63). Impβ is a spiral shaped molecule composed of 19 tandem helical hairpin motifs called HEAT repeats, with each hairpin comprising an outward-facing "A" helix and an inward-facing "B" helix. RanGTP and the cargo bind on the inner concave surface of the spiral, with different cargos interacting with different subsets of the HEAT repeats. The outer convex surface of Impβ interacts with nucleoporins that constitute the nuclear pore complex (54, 55, 56).

Structures are known for HIV-1 Rev and Rev-derived peptides, either alone (64, 65, 66, 67, 68), bound to an antibody (69), or in complex with RNA (70, 71, 72, 73, 74, 75) or with CRM1 and RanGTP (76). Standard lab isolates of Rev contain 116 amino acid residues. These define an N-terminal domain (NTD) that mediates nuclear import, RNA-binding, and multimerization and an intrinsically disordered C-terminal domain (CTD) responsible for nuclear export (Fig 1B). The CTD contains a leucine-rich nuclear export signal (NES; res. 75–83) which is recognized in the nucleus by CRM1 in complex with RanGTP. Residues C-terminal to the NES enhance protein stability (77, 78), become partly ordered when Rev forms filaments (67), and have

been predicted to interact with the NTD (79) and to regulate accessibility of the NES (80). The NTD includes an α-helical hairpin (res. 10–65) characterized by an arginine-rich motif (ARM; res. 35–50) that mediates RRE binding (13, 81, 82, 83, 84) and contains the NLS recognized by Impβ (42, 43). The N-terminal 10 residues are important for protein stability (78) but appear not to constitute a significant interaction epitope (31). Hydrophobic residues on either side of the ARM comprise an oligomerization domain responsible for Rev multimerization (13, 14, 85). These residues localize to both faces of the helical hairpin (denoted "head" and "tail" surfaces) and mediate three types of homotypic interactions (Fig 1C). Tail-to-tail (also called B–B) interactions mediate dimer formation after the initial association of Rev with the RRE (65, 74, 85). Head-to-head (or A–A) interactions mediate higher order Rev oligomer formation (69, 85). The third type of interaction, called C–C, involves the proline-rich interhelical loop and is proposed to bridge independently bound Rev dimers on the RRE surface (67). These three types of interface allow Rev to adapt to a wide variety of RNA sites on the RRE.

To gain insights into how Impβ associates with Rev, we analyzed the Impβ/Rev complex using diverse biochemical, mutational, and biophysical approaches together with molecular docking simulations. Unexpectedly, our data reveal that Impβ associates with two Rev monomers through distinct binding sites. Moreover, they allow us to localize one monomer near the N-terminal end of the

HEAT-repeat superhelix and to deduce an approximate structural model of the other Rev monomer bound near the opposite, C-terminal end. The ability to engage two independent binding sites on Impβ identifies Rev as an atypical nuclear import cargo.

# Results

### Rev forms oligomers that are disrupted by point mutations V16D and I55N

Our initial efforts to study the Impβ/Rev interaction were hampered by the tendency of Rev to aggregate under the low to medium ionic strength conditions required for stable Impβ/Rev complex formation. Previous studies reported several point mutations that hinder Rev multimerization (13, 65, 85, 86, 87, 88, 89), including mutations V16D and I55N which destabilize the homotypic A–A and B–B interfaces, respectively (Fig 1C) (85, 89). In particular, the double V16D/I55N mutant was previously used for biophysical studies of Rev, including single-molecule fluorescence spectroscopy (90) and NMR (66) experiments. Accordingly, we generated Rev mutants V16D and V16D/I55N, as well as a truncated version (residues 4–69) of the V16D/I55N mutant lacking the C-terminal domain. Bacterially expressed Rev proteins were purified to homogeneity, and their oligomeric state verified by size-exclusion chromatography/multi-angle laser light scattering (SEC/MALLS) in a high salt buffer (HSB) required to keep Rev soluble (Fig 2A). Wild-type (WT) Rev was eluted as two separate peaks: the first contained molecular species ranging between 330 and 500 kDa in mass (suggesting Rev multimers comprising ~24–36 monomers), whereas the second exhibited masses of 40–80 kDa (consistent with species comprising 3–6 Rev monomers), confirming the highly polydisperse nature of Rev. The V16D mutant was eluted later than WT Rev, yielding a broad peak with a mass distribution (15–45 kDa) consistent with a mixture of species comprising 1–3 Rev monomers. The mass observed at the maximum of this peak (~23 kDa) indicated a prevalence of the dimeric form (27.6 kDa), in agreement with previous findings (85, 89). By contrast, both the full-length and truncated versions of the V16D/I55N mutant eluted as relatively narrow peaks with mass distributions centered around 13.3 and 7.5 kDa, respectively, consistent with a monomeric state (13.8 and 8.0 kDa, respectively). The monodisperse nature of the V16D/I55N mutant facilitated several of the interaction studies with Impβ (isothermal titration calorimetry [ITC], NMR, and SAXS experiments) described below. For convenience, we refer to the full-length and truncated versions of this mutant as Rev[OD] (for oligomerization-deficient Rev) and Rev[OD]Δ, respectively.

### Impβ binds up to two monomers of Rev

We next assessed the ability of Rev to associate with Impβ. SEC analysis revealed that unbound Impβ eluted as a single peak, whereas preincubating Impβ with a molar excess of Rev[OD] led to the co-elution of both proteins within an earlier peak, indicating complex formation (Fig 2B). SEC analysis confirmed that Impβ also co-eluted with WT Rev, even though WT Rev was poorly soluble in the ionic strength conditions used for this experiment (Fig 2C). Impβ/Rev complex formation was confirmed by native gel analysis,

which revealed a single band for unbound Impβ and a discrete shift upon the addition of increasing concentrations of either WT Rev or Rev[OD] (Fig 2D, complex 1). Surprisingly, higher Rev concentrations resulted in a supershift, suggesting the formation of an additional species containing two or more Rev monomers (Fig 2D, complex 2). The similar results obtained with WT Rev and Rev[OD] suggest that residues V16 and I55 are not critical for the stability of the observed Impβ/Rev complexes. SEC/MALLS analysis also revealed molecular masses for Impβ/Rev[WT] complexes consistent with 1:1 and 1:2 stoichiometry (Fig 2E). A glutaraldehyde cross-linking experiment followed by mass spectrometry (MS) analysis corroborated the ability of Impβ to bind two Rev[OD] monomers (Fig S1A and B).

To further verify the stoichiometry of the Impβ/Rev association we performed native MS, which preserves non-covalent interactions and allows the mass of intact macromolecular complexes to be determined (91). To ensure correct interpretation of native MS spectra, we first used liquid chromatography coupled to electrospray ionization MS (LC/ESI-MS) to measure accurate masses under denaturing conditions for the unbound Impβ and Rev[OD] proteins; these were 97,300 and 13,264 Da, respectively, which are within 1 Da of the theoretical masses (Fig S2 and Table S1). Native MS analysis of unbound Impβ yielded a clear set of peaks whose $m/z$ values correspond to a mass of 97,296 Da (Fig 3A and B), closely matching that determined by ESI-TOF. We next incubated Impβ with Rev[OD] and measured the masses of the resulting species (Fig 3A, C, and D). Mixing Impβ with a twofold molar equivalent of Rev[OD] led to the appearance of two additional sets of peaks: the first corresponds to a mass of 110,558 Da (Fig 3C, blue circles), matching the expected mass of an Impβ/Rev[OD] heterodimer (110,563 Da), and the second to a mass of 123,829 Da (Fig 3C, green circles), consistent with a complex of Impβ bound to two Rev[OD] monomers (123,827 Da). Mixing Impβ with Rev[OD] in a 1:5 M ratio enhanced the intensity of this second set of signals (Fig 3D). Mixing Impβ with the truncated construct (Rev[OD]Δ) instead of full length Rev[OD] yielded an analogous set of peaks for Impβ/Rev[OD]Δ complexes having a stoichiometry of 1:1 (mass: 105,330 Da) and 1:2 (113,366 Da) (Fig 3A and E). Taken together, these results confirm the ability of Impβ to bind up to two monomers of Rev[OD].

We then investigated complex formation between Impβ and Rev[WT]. We overcame the tendency of Rev[WT] to oligomerize by exchanging its buffer into a high concentration of ammonium acetate before incubating with Impβ and using relatively low Rev[WT] concentrations (10–20 μM). Upon mixing Impβ with a twofold molar equivalent of Rev[WT], we detected peaks consistent with the formation of a 1:1 Impβ:Rev[WT] complex (110,476 Da; Fig 3A and F). With a fivefold molar equivalent of Rev[WT], we detected both 1:1 and 1:2 complexes (123,649 Da; Fig 3A and G). This confirms that Impβ can bind to either one or two molecules of WT Rev. Gel shift experiments (performed with either Rev[OD] or Rev[WT]) showed that RanGTP disrupted the formation of both the 1:1 and 1:2 complexes, indicating that RanGTP competes with both Rev monomers for binding to Impβ (Fig S3).

### Impβ recognizes the Rev helical hairpin through a high- and a low-affinity binding site

We next characterized the Impβ/Rev interaction using ITC. We recorded ITC profiles for the binding of Impβ to either Rev[OD] or Rev[OD]Δ because poor solubility and the tendency to multimerize

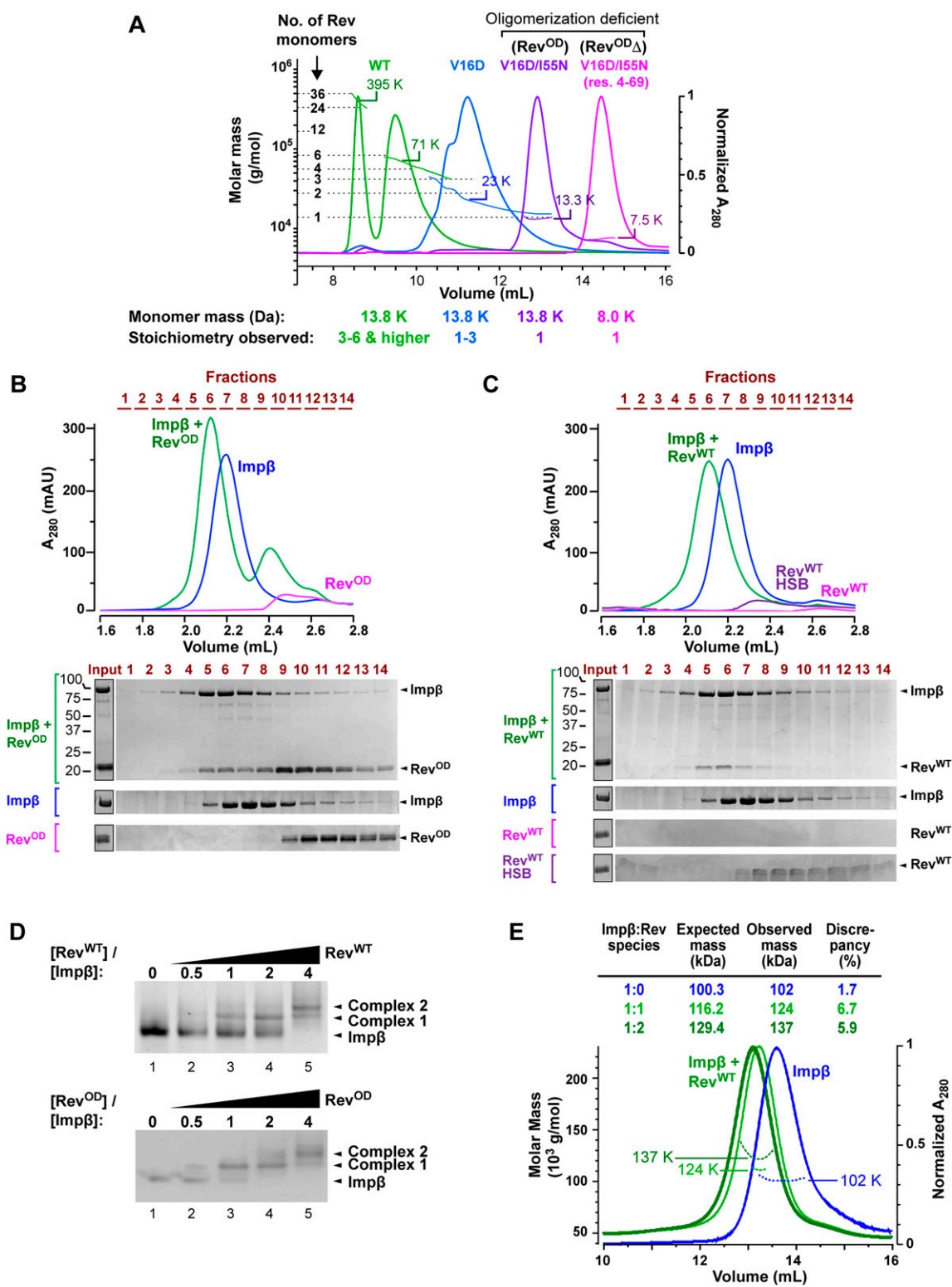

**Figure 2. HIV-1 Rev forms a stable complex with Impβ.**
**(A)** Analysis of different Rev constructs by size-exclusion chromatography/multi-angle laser light scattering (SEC/MALLS) using a Superdex 75 10/300 GL column. Elution curves recorded at 280 nm and molar mass distributions determined by MALLS are shown for the following Rev constructs: WT (green), V16D (blue), V16D/I55N (purple), and Rev4-69(V16D/I55N) (pink). The molar masses detected at the elution peaks are indicated. **(B)** SEC analysis of an Impβ/RevOD complex. *Top*: Elution profiles of samples containing Impβ (blue), RevOD (magenta), or a mixture of Impβ and RevOD (green). Fractions collected are indicated in brown. Chromatography was performed using a Superdex 200 5/150 Increase GL column. *Bottom*: SDS–PAGE analysis of the indicated fractions. **(C)** SEC analysis of an Impβ/RevWT complex. Elution profiles are shown for

precluded the use of WT Rev (Figs 4A and B and S4). Fitting the resulting binding isotherms using a model consisting of a single class of binding sites resulted in a significantly poorer fit compared with a model consisting of two nonsymmetric classes of binding sites, further confirming the ability of Impβ to bind two Rev monomers (Fig S5). (We note that the uniphasic, rather than bi-phasic, appearance of the observed binding isotherms is com-patible with the presence of two binding sites, as has been previously shown (92, 93)). These analyses yielded a dissociation constant ($K_d$) for the Impβ/Rev$^{OD}$ interaction of ~0.6 µM for the first site and 5 µM for the second site. Comparable $K_d$ values (0.6 and 8 µM) were obtained for the Impβ/Rev$^{OD}$Δ interaction, suggesting that the C-terminal domain missing from Rev$^{OD}$Δ does not con-tribute significantly to binding affinity.

The finding that Impβ binds one Rev monomer with sub-micromolar affinity and the other with ~10-fold weaker affinity is consistent with the sequential appearance of complexes observed when Impβ is titrated with Rev (Figs 2D and E and S1A). The interaction of Impβ with both the first and second Rev monomers is enthalpically driven (Fig 4C and D), suggesting electrostatic and hydrogen bond interactions that presumably take place between the highly basic ARM of Rev and the acidic cargo-binding inner surface of Impβ. Interest-ingly, although the higher affinity interaction (site 1) is associated with a negligible entropy change, the lower affinity interaction (site 2) is characterized by an unfavorable entropy term, suggesting that the binding of the second Rev monomer might significantly reduce the conformational entropy of the Impβ HEAT-repeat array, which is dy-namically highly flexible in the unbound state (94).

To further probe the regions of Rev involved in Impβ binding, we performed NMR spectroscopy of $^{15}$N-labeled Rev$^{OD}$ in the presence and absence of unlabeled Impβ. Triple resonance NMR was used to assign the backbone resonances of $^{15}$N-labeled Rev$^{OD}$. The $^1$H-$^{15}$N-BEST-TROSY spectrum of free Rev$^{OD}$ shows peaks concentrated between 7.5 and 8.5 ppm (Fig 5A), in agreement with a previous study (66) and consistent with the expectation that Rev contains many intrinsically disordered residues (95). In total, 67 of the 116 residues of Rev could be assigned. Assignment was hampered by resonance overlap caused by the high (14%) arginine composition of the Rev sequence and the large number (10) of prolines, as well as severe line-broadening in the C-terminus of the helical hairpin, possibly because of residual oligomerization at concentrations used for NMR measurement (90 µM). Analysis of the secondary $^{13}$C chemical shifts showed a strong helical propensity for residues within the helical hairpin domain, as expected, and significant helical propensity for residues 84–93 (Fig 5C). The latter residues are im-mediately downstream of the NES and their helical propensity might facilitate recruitment of the NES to the binding surface of CRM1.

The addition of Impβ to Rev$^{OD}$ had a dramatic effect on the spectrum, causing 28 peaks to disappear either completely (24 peaks) or nearly completely (4 peaks; >95% reduction in peak height) (Fig 5B and D). These peaks include all assigned peaks between residues 10 and 65, which delimit the helical hairpin domain. By contrast, peaks corresponding to N- and C-terminal regions of Rev (residues 2–9 and 73–116) were less affected by the presence of Impβ. This finding indicates that Impβ recognizes Rev primarily through its helical hairpin domain and interacts only weakly or not at all with the N-terminal extension and C-terminal domain, in agreement with the similar binding affinities measured by ITC for the interaction of Impβ with Rev$^{OD}$ and Rev$^{OD}$Δ.

### Impβ retains an extended conformation upon Rev binding

We next sought structural information on the Impβ/Rev complex using small-angle X-ray scattering, which was coupled with size-exclusion chromatography (SEC-SAXS) to ensure sample mono-dispersity. We measured data from Impβ alone or in complex with Rev$^{OD}$Δ and compared the observed scattering to that predicted for cargo-bound Impβ conformations (cargo coordinates removed) available from the Protein Data Bank (PDB) (Fig 6A). The scattering profile of unbound Impβ exhibits a shoulder at s ≈ 0.1 Å$^{-1}$ (Fig 6B), in agreement with previous studies of unbound murine and fungal Impβ proteins (49, 50, 97). The radius of gyration ($R_g$) and maximum dimension ($D_{max}$) of unbound Impβ determined from the pair distance distribution function were 37 and 125 Å, respectively, in-dicating that Impβ adopts a more extended conformation in so-lution compared with cargo-bound crystal structures (Fig 6A and C). Consistent with this finding, the observed scattering showed the best agreement ($χ^2$ values < 4) with scattering curves calculated for Impβ coordinates from PDB structures 1UKL and 3W5K, which ex-hibit the most elongated cargo-bound conformations (Fig 6C).

The binding of Rev$^{OD}$Δ caused only modest changes in the scattering profile relative to that of unbound Impβ (Fig 6B), cor-responding to a small decrease in $R_g$ and $D_{max}$ values (to 36.7 and 120 Å, respectively). These changes are consistent with the expected binding of Rev$^{OD}$Δ to the concave inner surface of the HEAT-repeat array, which appears not to undergo a dramatic conformational change relative to the unbound state. The small mass of Rev$^{OD}$Δ relative to Impβ and the large impact that changes in HEAT repeat conformation have on the predicted scattering curve preclude us from deducing a reliable structural model for the Impβ:Rev$^{OD}$Δ complex. Nevertheless, the data reveal that the complex retains an extended conformation resembling that of unbound Impβ and of the cargo-bound conformations 1UKL and 3W5K.

### Rev binding sites localize to N- and C-terminal regions of Impβ

To obtain additional structural insights into how Impβ recognizes Rev, we performed cross-linking–mass spectrometry (XL-MS) on the

---

Impβ (blue), Rev$^{WT}$ (magenta) and a mixture of Impβ and Rev$^{WT}$ (green). Because most free Rev$^{WT}$ forms insoluble aggregates in the buffer conditions used and hence is not detected in the elution, the elution profile of Rev$^{WT}$ analyzed on a column pre-equilibrated in high-salt buffer was included as an additional reference (purple). **(D)** Native gel analysis showing the association of Impβ with one or more monomers of Rev$^{WT}$ (top) and Rev$^{OD}$ (bottom). **(E)** SEC/MALLS analysis of Impβ in the presence and absence of Rev. Elution curves recorded at 280 nm and molar mass distributions determined by MALLS are shown for His-tagged Impβ in the absence (blue) and presence of a either a twofold (light green) or fourfold (dark green) excess of Rev$^{WT}$. The observed masses of 124 and 137 kDa are consistent with those expected for 1:1 (116.2 kDa) and 1:2 (129.4 kDa) Impβ:Rev stoichiometry. Chromatography was performed using a Superdex 200 10/30 GL column. Injected concentrations were 20 µM Impβ and 40 or 80 µM Rev$^{WT}$.

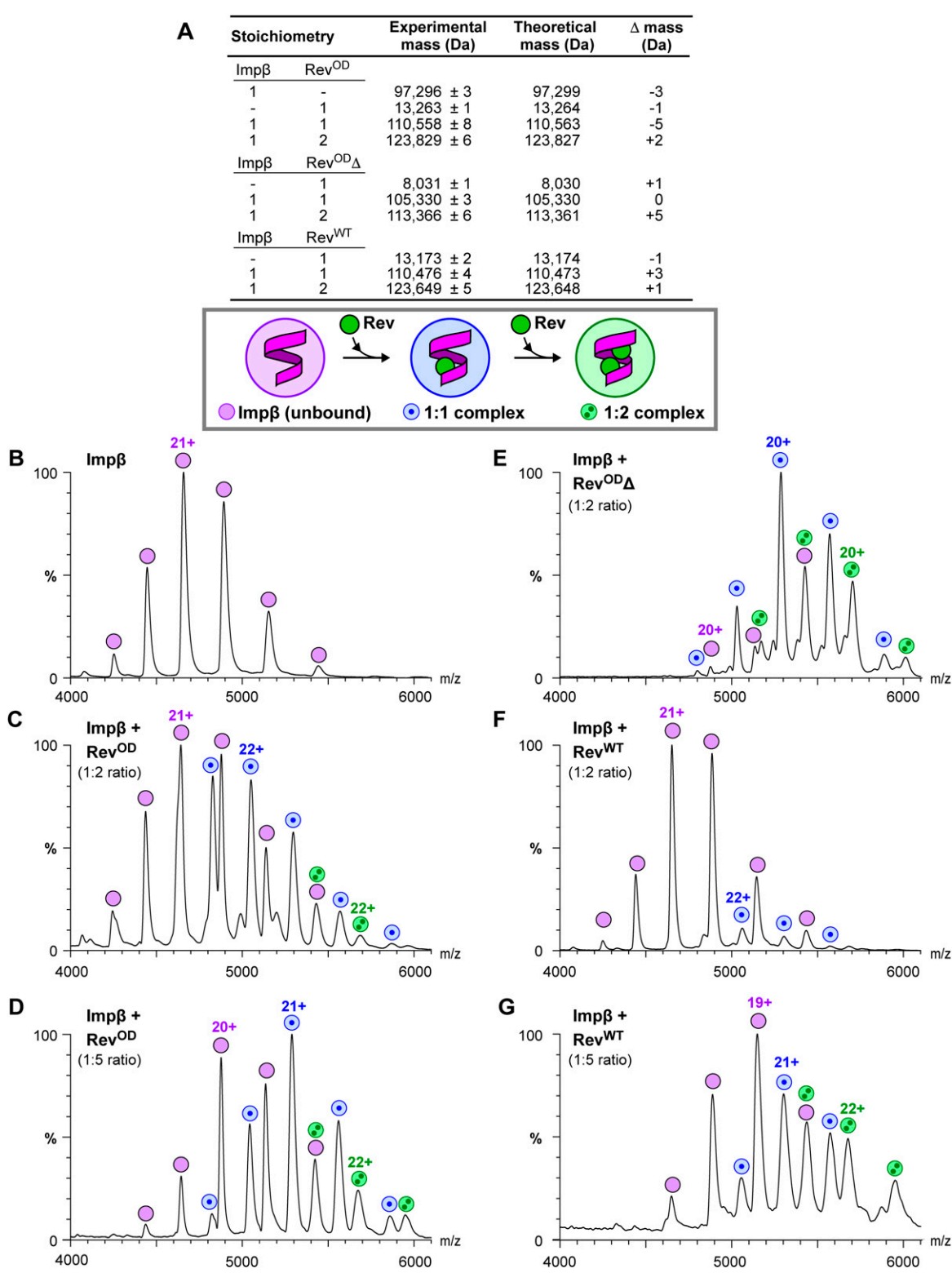

**Figure 3. Native MS reveals that Impβ binds up to two Rev monomers.**
**(A)** Summary of masses observed by native MS. **(B, C, D, E, F, G)** Native MS spectra. Peaks labeled by magenta circles correspond to unbound Impβ. Peaks labeled by blue circles with a single dot or by green circles with two dots correspond to Impβ/Rev complexes with 1:1 or 1:2 stoichiometry, respectively. **(B)** Spectrum of Impβ in the absence of Rev. **(C, D)** Spectra of Impβ incubated with a (C) twofold or (D) fivefold molar equivalent of Rev[OD]. **(E)** Spectra of Impβ in the presence of the truncated construct Rev[OD]Δ. **(F, G)** Spectra of Impβ incubated with a (F) twofold or (G) fivefold molar equivalent of Rev[WT].

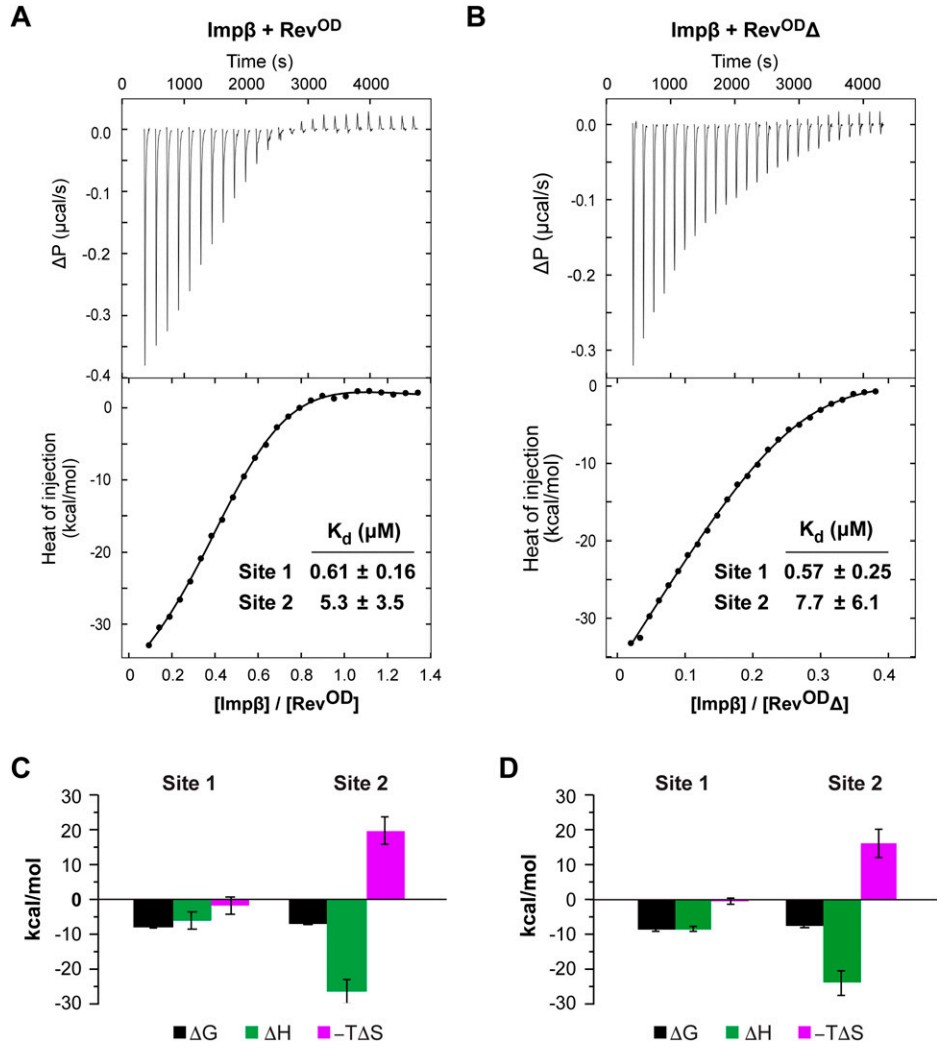

**Figure 4. Rev binds Impβ at two sites with sub- and low micromolar affinity.**
**(A, B)** Representative isothermal titration calorimetry profiles of the binding of Impβ to (A) Rev[OD] and (B) Rev[OD]Δ. *Top*: Differential power time course of raw injection heats for a titration of Impβ into the Rev protein solutions. *Bottom*: Normalized binding isotherms corrected for the heat of dilution of Impβ into buffer. The solid line represents a nonlinear least squares fit using a model consisting of two nonsymmetric classes of binding sites. **(C, D)** Thermodynamic values obtained from isothermal titration calorimetry data for the binding of Impβ to (C) Rev[OD] and (D) Rev[OD]Δ. Binding to site 1 is characterized by a favorable enthalpy ($\Delta H \approx -8$ kcal/mol) and a negligible entropy change for both Rev[OD] and Rev[OD]Δ, whereas binding to site 2 is associated with an unfavorable entropy change that is offset by a large negative enthalpy change. Thermodynamic parameters (in kcal/mol) are as follows: Rev[OD]: $\Delta G = -8.3 \pm 0.1$ and $-7.2 \pm 0.4$, $\Delta H = -6.3 \pm 2.4$ and $-26.5 \pm 3.5$, $-T\Delta S = -2.1 \pm 2.5$ and $19.3 \pm 3.9$ for sites 1 and 2, respectively; Rev[OD]Δ: $\Delta G = -8.5 \pm 0.2$ and $-7.2 \pm 0.8$, $\Delta H = -8.3 \pm 0.6$ and $-23.6 \pm 3.6$, $-T\Delta S = 0.3 \pm 0.8$ and $16.5 \pm 4.0$ for sites 1 and 2, respectively. Data represent the mean ± SD from three independent replicates. All replicate profiles are shown in Fig S4.

Impβ/Rev complex using the amine-reactive cross-linker bissulfosuccinimidyl suberate (BS3). Although Rev contains two lysine residues (Lys20 and Lys115), BS3-mediated cross-links with Impβ were only detected for Lys20 and for the Rev N-terminal amino group (Fig 7A). Conversely, peptides containing BS3-modified Lys residues were identified for 28 of the 43 lysines present in Impβ. These residues are distributed over the length of Impβ and can be classified into three groups. Group 1 formed cross-links with both the N-terminus and Lys20 of Rev; group 2 formed cross-links with the flexible Rev N-terminus but not with Lys20; and group 3 formed monolinks (where only one end of BS3 was covalently bound to the protein) or cross-links with other Impβ residues but did not form detectable cross-links with Rev. Because Lys20 is located on Rev helix α1, group-1 lysines on Impβ are expected to localize close to Rev's helical hairpin, whereas lysines in groups 2 and 3 are expected to be farther away. Interestingly, although group-3 residues primarily reside on the A helix of a HEAT repeat and localize to the outer convex surface of Impβ, nearly all group-1 residues reside on a B helix and localize to the inner concave surface (Fig 7B and C), consistent with Rev recognition by the inner surface of Impβ.

The 11.4 Å linker arm length of BS3 allows the Cα atoms of cross-linked lysines to be up to ~24 Å apart; however, to allow for conformational dynamics, distance constraints of 30–35 Å are typically used for the purposes of 3D structural modeling (98, 99, 100). For such modeling, the solvent-accessible surface (SAS) distance between cross-linked Lys residues is more informative than the straight-line (Euclidean) distance separating them as the latter may trace a path sterically inaccessible to the cross-linker (101). Group 1 includes a cluster of lysines within HEAT repeats 1 and 2 and a second cluster within repeat 19 at the opposite extremity of Impβ. These two clusters are too far apart to form cross-links to the same Rev K20 residue position (Fig 7D). For example, if one considers the Impβ conformation shown in Fig 7E (from PDB 1UKL), the Cα atoms of residues Lys23 and Lys873 are separated by a SAS distance of 103 Å (Euclidean distance of 91.4 Å). Even for the most compact known conformation of Impβ (from PDB 3LWW), residues Lys859, Lys867, and Lys873 are separated from Lys23 by a SAS distance of over 70 Å, twice the BS3 cross-linking distance constraint (Fig S6A and B). These findings are consistent with the ability of Impβ to bind to two Rev molecules and suggest that the

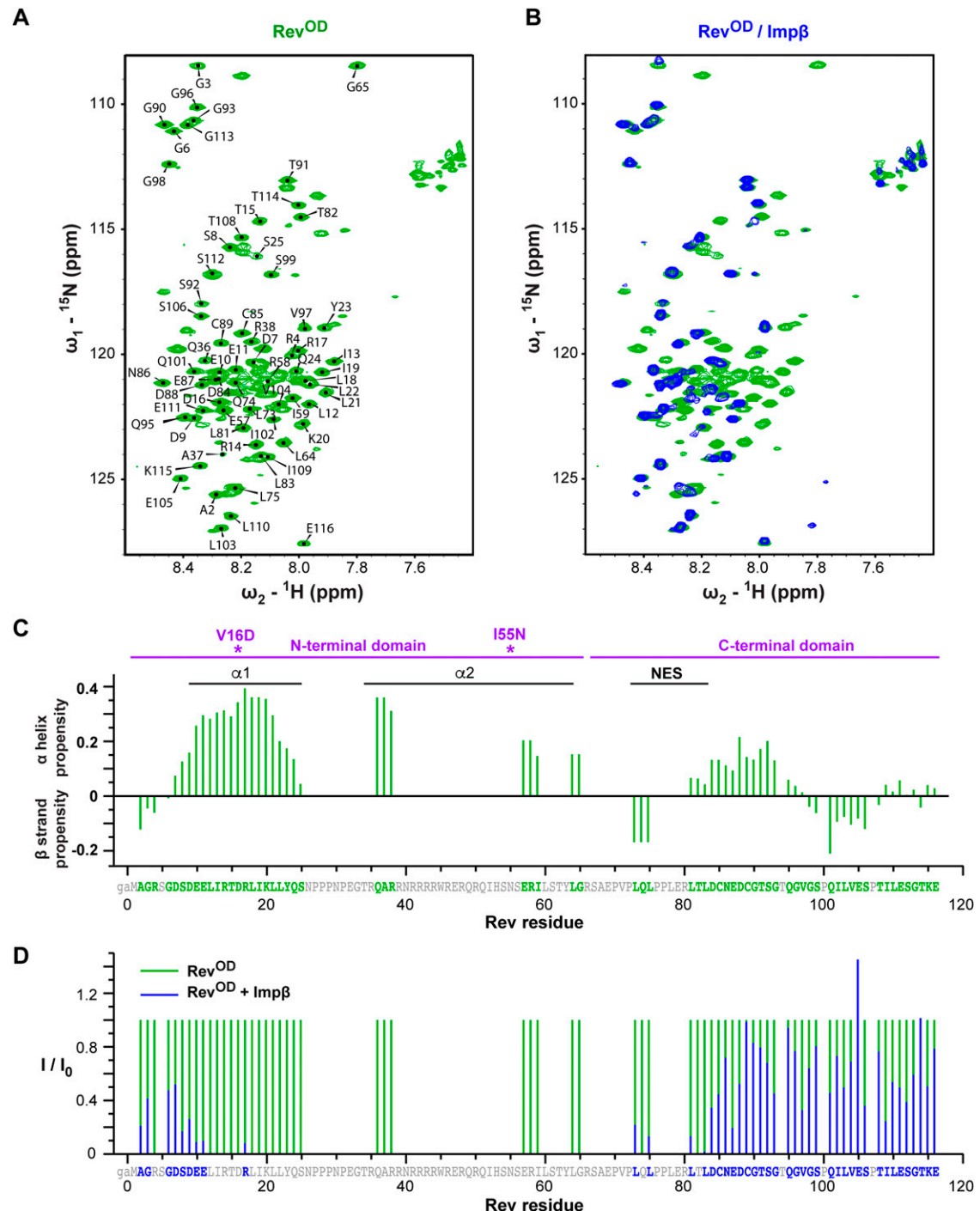

**Figure 5. NMR analysis of Rev binding by Impβ.**
(A) $^1$H,$^{15}$N HSQC spectrum of free Rev$^{OD}$ (600 MHz, 283K). (B) $^1$H,$^{15}$N-HSQC spectrum Impβ-bound Rev$^{OD}$ (blue) superimposed on that of unbound Rev$^{OD}$ (green). (C) Secondary chemical shifts (96) of unbound Rev$^{OD}$. The two point mutations are indicated by asterisks. (D) Ratio of peak intensities for Rev residues in the presence and absence of Impβ (600 MHz, 283K).

two clusters of group-1 lysines form cross-links with distinct Rev monomers, one bound close to the N-terminal end of Impβ and the other close to the C-terminal end. We refer hereafter to these two Rev binding sites on Impβ as the N-site and C-site, respectively.

Applying a 35 Å distance constraint from each residue in the two group-1 lysine clusters localizes the Lys20 Cα atoms of the two Rev monomers within an ellipsoidal volume next to the N- and C-terminal HEAT repeats with a positional uncertainty of ~22 Å (Figs 7F and S6C). In contrast, group-1 residues Lys206 and Lys537 could

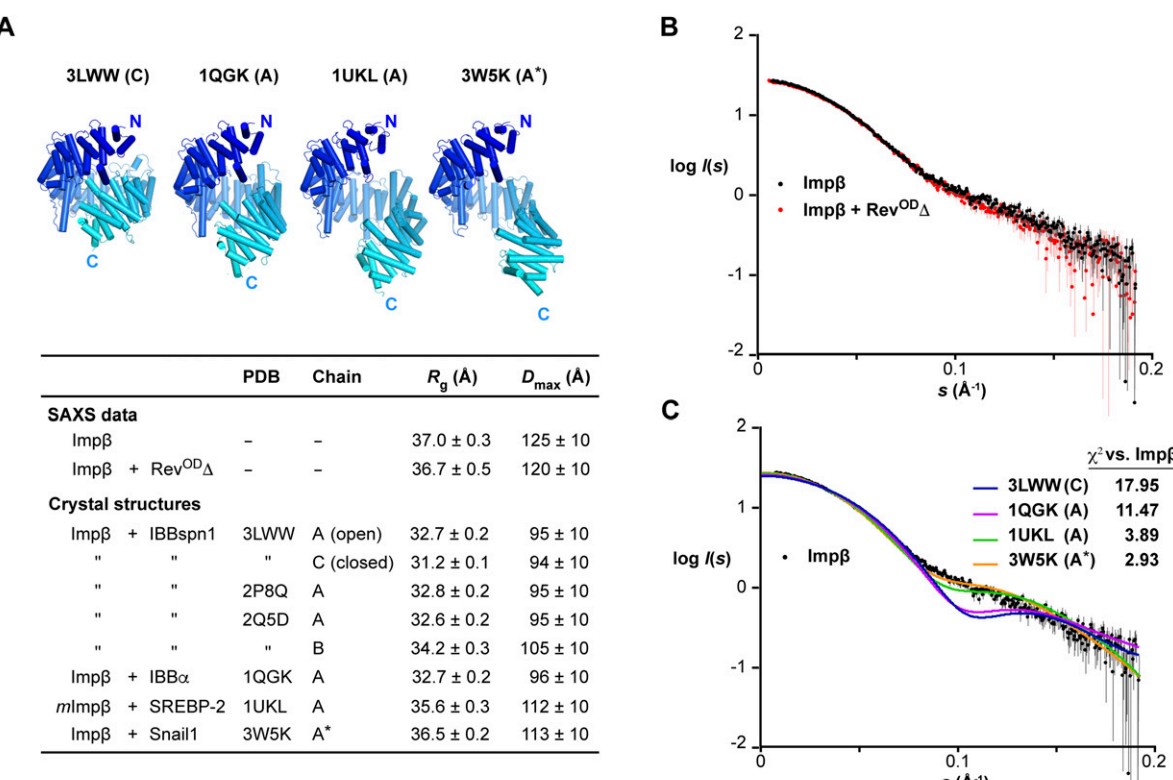

**Figure 6. Imp$\beta$ retains an extended conformation upon binding Rev.**
**(A)** *Top*: Representative crystal structures of Imp$\beta$ illustrating different degrees of compaction of the HEAT-repeat array. *Bottom*: Model-independent parameters obtained from SAXS data compared with values calculated for available Imp$\beta$ crystal structures using the programs CRYSOL (127) and PRIMUS (126). All structures are of human Imp$\beta$ except for 1UKL which is of murine Imp$\beta$. Chain A in 3W5K lacks coordinates for Imp$\beta$ residues 1–15. For the calculation of $R_g$, $D_{max}$, and $\chi^2$ values, this chain was extended to include these residues (denoted A*) by replacing HEAT repeat 1 (residues 1–31) by the corresponding residues from 1UKL following local alignment of the two structures. **(B)** Scattering data from unbound Imp$\beta$ (black) and the Imp$\beta$/Rev$^{OD}\Delta$ complex (red). **(C)** Scattering data from unbound Imp$\beta$ (black) compared with profiles calculated from representative cargo-bound conformations exhibiting different degrees of elongation (colored lines).

potentially cross-link with either Rev monomer. The fact that both these residues localize to the concave inner surface of Imp$\beta$ confirms that at least one of the two Rev monomers is bound to this surface.

### Imp$\beta$ recognizes peptides derived from Rev helix $\alpha$2

To delineate the region(s) of Rev recognized by Imp$\beta$, we examined the ability of different Rev-derived peptides to interact with Imp$\beta$ in a thermal shift assay. We used a set of 27 peptides (each comprising 15 residues) that span the entire Rev sequence, with an 11-residue overlap between consecutive peptides. We used differential scanning fluorimetry (DSF) to assess the ability of peptides to increase the thermal stability of Imp$\beta$. In the absence of peptides, the melting temperature ($T_m$) observed for Imp$\beta$ was 35.4 ± 0.7°C (Fig 8A). Although most Rev peptides did not significantly increase this value ($\Delta T_m < 0.5$°C), three consecutive Rev peptides (numbered 9 to 11) strongly stabilized Imp$\beta$ ($\Delta T_m > 2$°C), with peptide 10 ($^{37}$ARRNRRRRWRERQRQ$^{51}$), which contains all but one of the ARM's 10 Arg residues, yielding the strongest effect (Fig 8A and B). These peptides collectively span Rev residues 33–55 on helix $\alpha$2 and the $\alpha$1-$\alpha$2 loop and nearly perfectly match the ARM motif (res. 35–50) (Fig 8C).

The increased stabilization observed for peptide 10 compared with peptide 9 ($^{33}$GTRQARRNRRRRWRE$^{47}$) suggests that the Rev $^{48}$RQRQ$^{51}$ motif includes an important binding epitope. Similarly, although Imp$\beta$ was strongly stabilized by Rev peptide 11 ($^{41}$RRRRWRERQRQIRSI$^{55}$), little or no stabilization was observed with peptide 12 ($^{45}$WRERQRQIRSISGWI$^{59}$), indicating that the tetra-arginine motif ($^{41}$RRRR$^{44}$) is critical for Imp$\beta$ binding. Finally, peptide 8 ($^{29}$PSPEGTRQARRNRRR$^{43}$), which contains three of these four Arg residues and a total of six closely spaced Arg residues, only modestly stabilizes Imp$\beta$ ($\Delta T_m = 1.0 ± 0.7$°C), indicating that clustered Arg residues are not sufficient for high-affinity binding and that sequence context also plays an important role.

### The N-terminal tip of Rev helix $\alpha$2 is a major Imp$\beta$ binding epitope

We next performed mutagenesis experiments to identify individual Rev residues critical for Imp$\beta$ recognition. Because Rev binding is likely to involve a significant electrostatic component, we sought to destabilize the Imp$\beta$/Rev interface by introducing charge-reversal mutations on Rev. We first made mutants in which two or three basic residues were simultaneously replaced by Asp residues (mutants R1–R5; Fig 9A). We used a competitive fluorescence

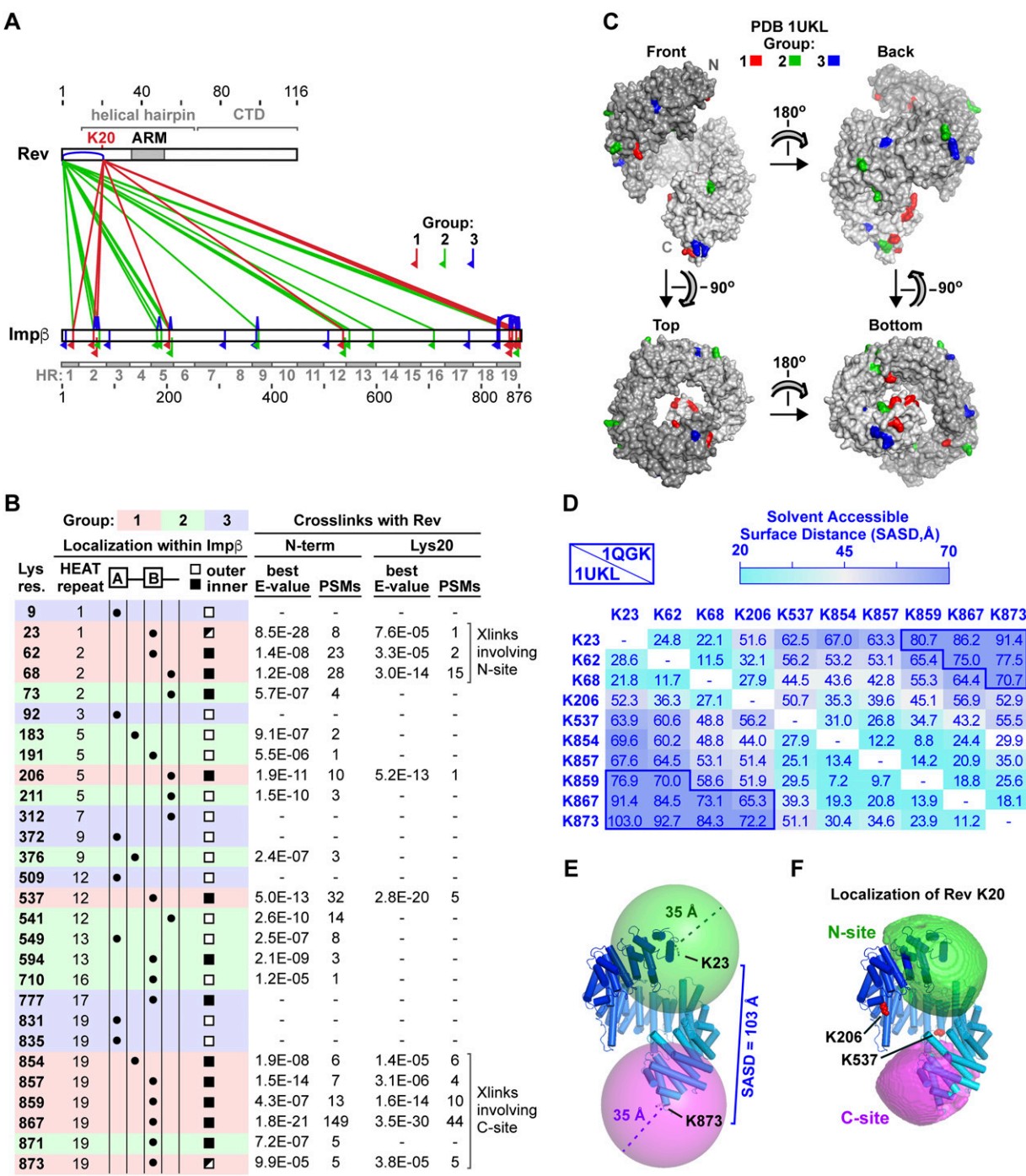

**Figure 7. Cross-linking-MS localizes two Rev binding regions on Impβ.**
**(A)** Graphical summary of cross-links. Impβ-Rev cross-links involving the N-terminus or Lys20 residue of Rev are shown in green and red, respectively. Rev–Rev and Impβ-Impβ cross-links and monolinks are shown in blue. Red, green, and blue inverted flags indicate Impβ residues that form cross-links with Rev Lys20 (group 1), with both Rev Lys20 and the Rev N-terminus (group 2), or for which no cross-links with Rev were detected (group 3), respectively. The 19 HEAT repeats (HR) of Impβ are indicated. **(B)** Impβ Lys residues modified by BS3 and detected in cross-links or monolinks. Group-1, -2, and -3 lysines are highlighted in red, green, and blue, respectively. For group-1 and -2 lysines, the number of peptide spectrum matches (PSMs) and the best pLink E-value score are indicated. **(C)** Surface representation of Impβ showing the location of cross-linked group-1, -2, and -3 Lys residues, colored red, green, and blue, respectively. Shading is from dark to light gray from N- to C-terminus. **(D)** Solvent-accessible surface distances (SASDs) between pairs of Impβ group-1 Lys residues. The upper and lower triangles show distances for the conformations of Impβ bound to the importin α IBB domain (PDB 1QGK) and to SREBP-2 (PDB 1UKL), respectively. SASDs over 70 Å are outlined in dark blue. Distances were calculated using the Jwalk webserver (101). The SAS distances shown here and in Fig S6B suggest that Lys23, Lys62, and Lys68 cross-link with Rev at the N-site and that Lys854, Lys857, Lys859, Lys867, and Lys873 cross-link with Rev at the C-site. **(E)** Spheres of radius 35 Å centered on the Cα atoms of group-1 residues Lys23 and Lys873 show that BS3 molecules bound to these two lysines cannot cross-link to the same Rev Lys20 position. **(F)** Localization of the Cα atom of Rev Lys20. Green and magenta volumes show the N- and C-terminal regions of space within cross-linking distance of group-1 residues in either HEAT repeats 1 and 2 (K23, K62, K68) or repeat 19 (K854, K857, K859, K867, and K873), respectively,

polarization (FP) assay to compare the ability of these proteins to bind Impβ and displace a peptide cargo. In this assay, the binding of Impβ to a fluorescently labeled Rev peptide (Rev-NLS) yields an FP signal that decreases when the peptide is displaced by the addition of unlabeled Rev. The half-maximal inhibitory concentration (IC$_{50}$) of the Rev protein tested (or its negative logarithm, $p$IC$_{50}$) provides an indirect measure of Impβ-binding affinity, with lower IC$_{50}$ (higher $p$IC$_{50}$) values corresponding to stronger binding.

Representative assays are shown in Fig 9B, and the results summarized in Fig 9C–E and Table S2. WT Rev competes with the Rev-NLS peptide with an IC$_{50}$ value of 0.5 $\mu$M, comparable to the $K_d$ value of 0.6 $\mu$M obtained by ITC. Of the five mutants tested, three (R2-R4) yielded a large (10-fold) increase in IC$_{50}$, whereas the other two (R1, R5) gave more modest (3- and 1.7-fold) increases. The three mutants with the greatest effect localized to one end of the helical hairpin (the N-terminal half of helix α2 and the interhelical loop) and involved Arg clusters on both the A- and B-faces of Rev, whereas the mutants with more modest effects localized to the opposite end of the domain (helix α1 and the C-terminal half of helix α2) (Fig 9A), in line with our peptide-scanning data.

We then generated single-point mutations of the nine Arg residues altered in mutants R2-R4. Four of these (R35D, R38D, R39D, and R41D) significantly compromised the ability of Rev to compete with the Rev-NLS peptide, increasing the IC$_{50}$ by a factor of 1.6-1.9 (versus a factor of 1.1–1.3 for the other mutants), corresponding to a drop in $p$IC$_{50}$ of 0.20–0.27 (versus 0.04–0.11) (Fig 9D and E). Interestingly, the sum of the Δ$p$IC$_{50}$ values (−0.72) for mutations R35D, R38D, and R39D approached that observed for the corresponding R2 mutant (−1.0), indicating that the effect of the triple mutation primarily reflects the additive effects of the single mutations (Fig 9E). In contrast, the Δ$p$IC$_{50}$ values for triple mutants R3 and R4 were three to four times larger in magnitude than the summed Δ$p$IC$_{50}$ values of the corresponding single mutations, revealing that reversing the charge at these combined positions had a synergistic effect on disrupting binding. Notably, plotting the Δ$p$IC$_{50}$ values of the nine R→D mutations onto the structure of Rev reveals a progressively greater effect of the mutation on Impβ binding as one proceeds along helix α2 toward its N-terminus and the interhelical loop (Fig 9F), consistent with the spatial trend noted above for the triple mutants. These findings point to the N-terminal tip of Rev helix α2 as a major binding epitope recognized by Impβ.

### Charge-reversal mutations on Impβ and Rev show compensatory effects

We next sought to identify Impβ residues implicated in Rev recognition by introducing charge-reversal mutations on the acidic inner surface of Impβ. Of the 57 Asp and Glu residues that localize to the protein's inner surface, we selected 22 that were prominently exposed or clustered into acidic patches for substitution by Arg residues. To enhance the potential disruptive effect on Rev binding, we initially introduced these substitutions as double, triple, or quadruple mutations (mutants B1-B7, Fig 10A). We then used our FP inhibition assay to compare the ability of WT Rev to displace the Rev-NLS peptide from WT or mutant forms of Impβ. Only minor changes in the IC$_{50}$ value of Rev were observed when WT Impβ (IC$_{50}$ = 0.5 $\mu$M) was replaced by mutants B1-B7 (IC$_{50}$ = 0.43–0.55 $\mu$M) (Fig 10B and Table S2), consistent with the expectation that the Impβ mutations should not significantly alter Rev's ability to compete with the Rev-NLS peptide. We then repeated the assay using Rev double and triple mutants R1-R5. Strikingly, although the IC$_{50}$ values of mutants R1 and R5 showed little variation in assays performed with WT or mutant Impβ, the IC$_{50}$ values of mutants R2-R4 varied considerably (Fig 10C and Table S2). Most notably, these values decreased from ~5 $\mu$M when tested against WT Impβ to ~1.6 $\mu$M against mutant B2, signifying that Rev mutants R2-R4 competed with the Rev-NLS peptide with ~3 times greater efficacy when WT Impβ was replaced by the B2 mutant. More modest (1.5–1.8-fold) decreases in IC$_{50}$ were also observed with mutants B1, B3, and B4. These findings reveal that the Asp/Glu→Arg substitutions on Impβ partly compensate for the adverse effects of the Rev Arg→Asp mutations on the ability of Impβ to bind Rev, suggesting a possible electrostatic interaction between the mutated acidic and basic residues (Fig 10D).

To assess the size of this compensatory effect and facilitate comparison across different combinations of Impβ and Rev mutants, we used the parameter ΔΔ$p$IC$_{50}$, whose derivation is illustrated for mutants B2 and R4 in Fig 10E. In this figure, replacing WT Rev (solid black circles) by mutant R4 (solid red circles) causes the $p$IC$_{50}$ to drop from 6.3 to 5.3, yielding a Δ$p$IC$_{50}$ value of −1, whereas replacing both WT Rev and WT Impβ by mutants R4 and B2, respectively (red diamonds), results in a Δ$p$IC$_{50}$ of only −0.5. Thus, compared with the former assay, the latter assay causes Δ$p$IC$_{50}$ to shift from −1 to −0.5, yielding a ΔΔ$p$IC$_{50}$ value of +0.5. The ΔΔ$p$IC$_{50}$ values for selected combinations of Impβ and Rev mutants are shown in Fig 10F and G.

To pinpoint potentially interacting Impβ and Rev residues, we generated single charge-reversal mutations of the nine Impβ residues concerned by mutants B2-B4 and tested these in FP inhibition assays against Rev triple mutants R2-R4 and the corresponding Rev single-point mutants shown in Fig 9D. The results revealed several combinations of Impβ and Rev mutants that gave notable compensatory effects (Figs 10F and G and S7). These effects were generally more significant when either Impβ or Rev contained multiple mutations, as observed for example between Rev mutants R2-R4 and Impβ mutant D288R, between Rev mutant R3 and Impβ mutant D339R, and between Rev mutant R48D and Impβ mutants B3 and B4. Of the 51 pairwise combinations of single-point mutants tested, the strongest compensatory effects were seen for Impβ mutant D288R with Rev mutants R42D and R46D (ΔΔ$p$IC$_{50}$ = 0.41 and 0.33, respectively) and to a lesser extent R43D (ΔΔ$p$IC$_{50}$ = 0.17) (Figs 10G and S7A–C). ΔΔ$p$IC$_{50}$ values exceeding 0.1 were also observed for Impβ mutant E299R with Rev mutants R42D, R43D, and R46D, for Impβ E289R with Rev R46D, and for Impβ E437R with Rev R48D (Figs 10G and S7D–G). These findings identify Impβ acidic residues on the B helices of HEAT repeats 7 and 10 that are likely to be spatially

defined by the intersection of 35 Å spheres centered on the Cα atoms of these residues. If the centroid of each envelope is used to estimate the position of the Rev K20 Cα atom, then the positional uncertainty, calculated as the rmsd of each grid point within the envelope (sampled on a 2 Å grid) relative to the centroid, is 22.8 Å for the N-site and 21.6 Å for the C-site for the 1UKL conformation. **(C, E, F)** The Impβ conformation shown in panels (C, E, F) is that of SREBP-2-bound Impβ (PDB 1UKL).

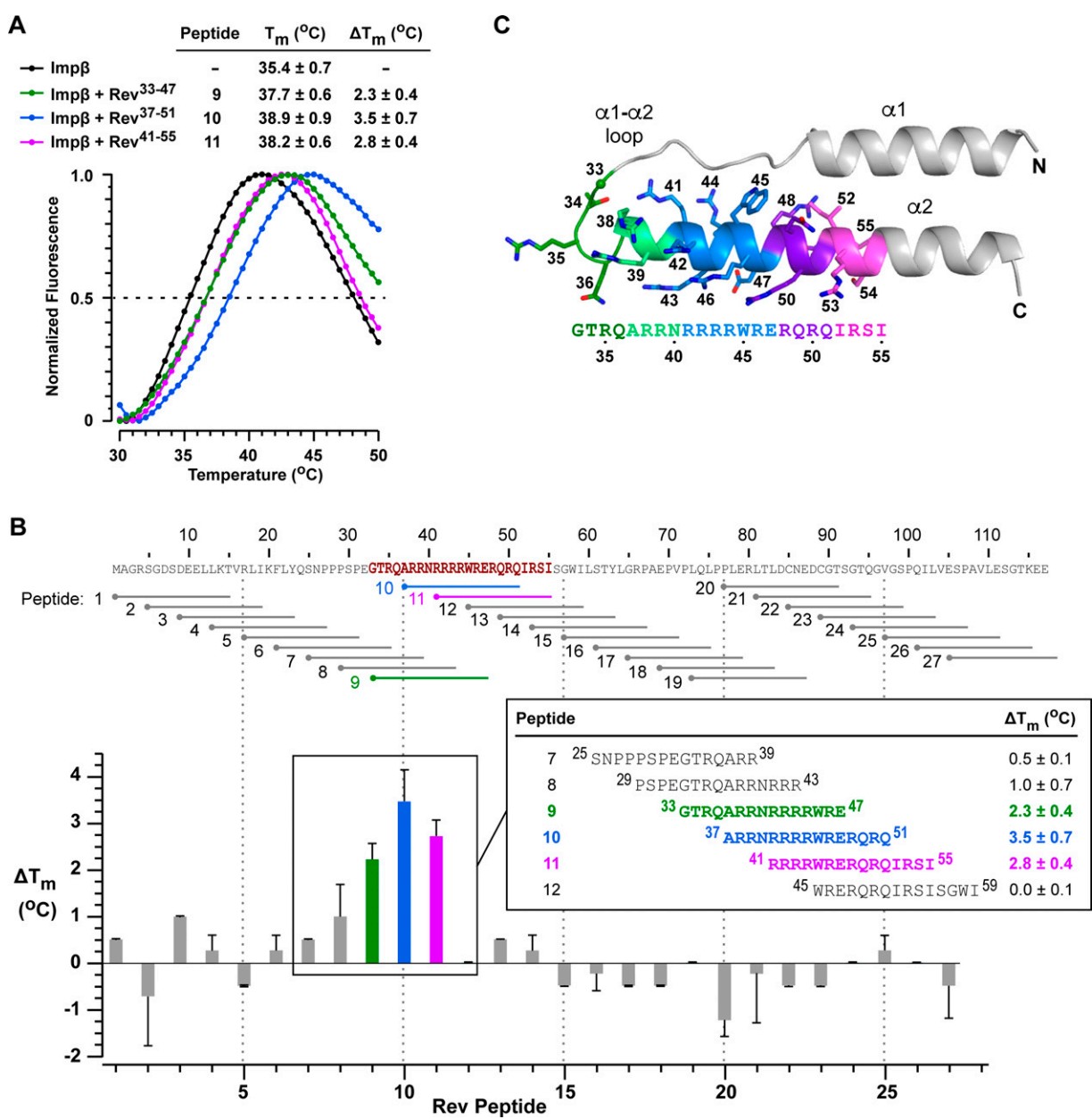

**Figure 8. Impβ recognizes Rev peptides derived from helix α2 and the α1-α2 loop.**
(A) Examples of thermal denaturation curves measured by differential scanning fluorimetry of Impβ in the presence and absence of Rev peptides. Data are shown for Rev peptides 9–11. The melting temperature ($T_m$) and difference in $T_m$ compared with unbound Impβ ($\Delta T_m$) are listed as mean values ± SD from three independent experiments. (B) Summary of $\Delta T_m$ values determined by differential scanning fluorimetry analysis of Impβ for all 27 peptides spanning the Rev sequence. Details are shown for peptides 7–12. (C) Structure of the Rev N-terminal domain highlighting the residues in helix α2 and the α1-α2 loop spanned by Rev peptides 9–11. Atomic coordinates are taken from PDB 2X7L (69).

proximal to basic residues on Rev helix α2 (Fig 10H), providing important clues into the relative binding orientations of Impβ and Rev.

## Molecular docking simulations associate certain compensatory mutations with the C-site of Impβ

The mutual proximity of Rev residues Arg42 and Arg46 on helix α2 (Fig 10H) and the short distances (<18 Å) separating Impβ residue

Asp288 from residues Glu299 and Glu437 relative to the width of the Rev helical hairpin (20–25 Å) make it likely that the compensatory effects summarized in Fig 10H are all mediated by the same Rev monomer. However, whether that Rev monomer localizes to the N- or C-site on Impβ is unclear. To explore this question, we performed molecular docking simulations to assess whether certain distance restraints could be attributed reliably to either site. We docked the Rev helical hairpin onto the Impβ surface (using the extended 1UKL conformation as suggested by our SAXS data) with the program

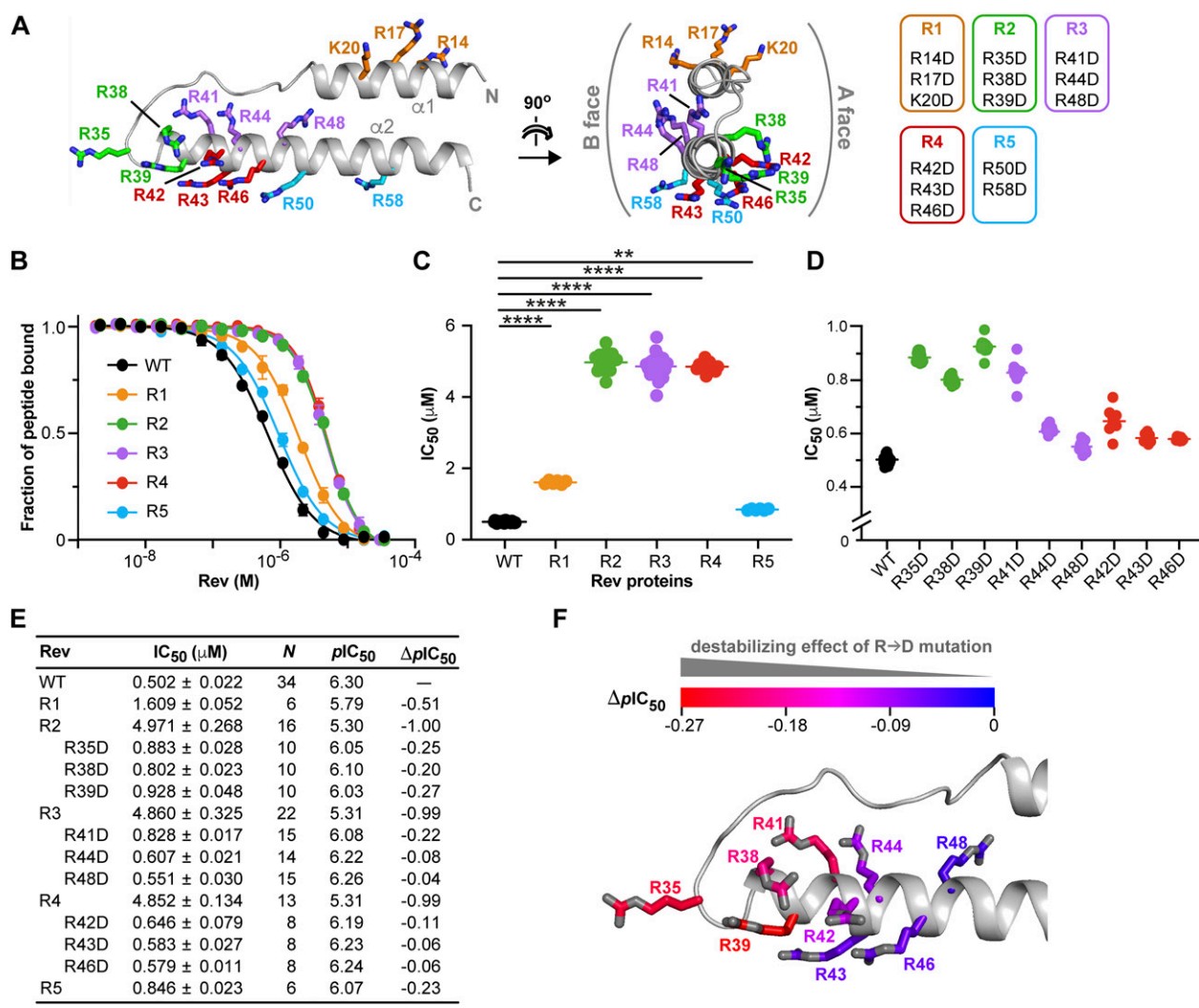

**Figure 9. Charge-reversal mutations identify an Impβ-binding epitope on Rev.**
**(A)** Rev double and triple R/K→D substitution mutations. **(B)** Competitive fluorescence polarization (FP) inhibition assays showing the ability of WT and mutant forms of Rev to displace a fluorescently labeled Rev-NLS peptide from Impβ. A representative experiment is shown for each protein. Data shown are mean and SD values from two technical replicates. **(C)** Plot of $IC_{50}$ values derived from FP inhibition assays for WT Rev and double and triple mutants. ****$P \leq 0.001$; **$P \leq 0.01$. $P$-values were determined using an ordinary ANOVA test. **(D)** Plot of $IC_{50}$ values for WT Rev and single R→D point mutants. **(E)** Summary of $IC_{50}$ (mean ± SD) and corresponding $pIC_{50}$ values. $N$ represents the number of biological replicates. $\Delta pIC_{50}$ values are reported relative to WT Rev. **(F)** View of the Rev helical hairpin showing the Arg residues selected for single-point mutations. Carbon atoms are colored from blue to red in order of increasing ability of the R→D substitution to compromise Impβ recognition, as measured by $\Delta pIC_{50}$ values.

HADDOCK, which exploits experimentally derived interaction restraints to guide the docking procedure (102). (For clarity, in the following description of interaction restraints, a superscript is used to indicate whether a residue belongs to either Rev or Impβ). In initial simulations, we specified distance restraints derived from compensatory mutations involving residue pairs Asp288$^{Impβ}$: Arg42$^{Rev}$ and Asp288$^{Impβ}$:Arg46$^{Rev}$ and combined these with BS3 cross-linking distance restraints for Rev bound to the N-site (experiment 1) or C-site (experiment 2) (Table S3A). Both simulations yielded similar scoring statistics (e.g., HADDOCK score, cluster size, buried surface area) for the top-ranked cluster of solutions (Table S3B). These solutions positioned Rev on the inner surface of Impβ next to HEAT repeats 1–8 (experiment 1) or repeats 7–19 (experiment 2), with Rev helix α2 oriented roughly anti-parallel or parallel,

respectively, to the Impβ superhelical axis (Fig S8A and B). (We refer hereafter to these two generic Rev orientations as "anti-parallel" or "parallel," respectively). In both experiments, most of the docking solutions closely resembled the top-ranked solution with only minor variations in Rev orientation, indicating a relatively limited region of configurational space consistent with the docking parameters used (Fig S9A and B).

The above experiments included only two of the eight possible distance restraints suggested by our mutagenesis data (Fig 10H) and in particular omitted the distance restraint between residues Glu437$^{Impβ}$ and Arg48$^{Rev}$. Strikingly, however, experiment 2 (with Rev in the C-site) yielded solutions in which these two residues were relatively close to each another, including the second ranked solution which positioned these residues <5 Å apart (Fig S8B). In

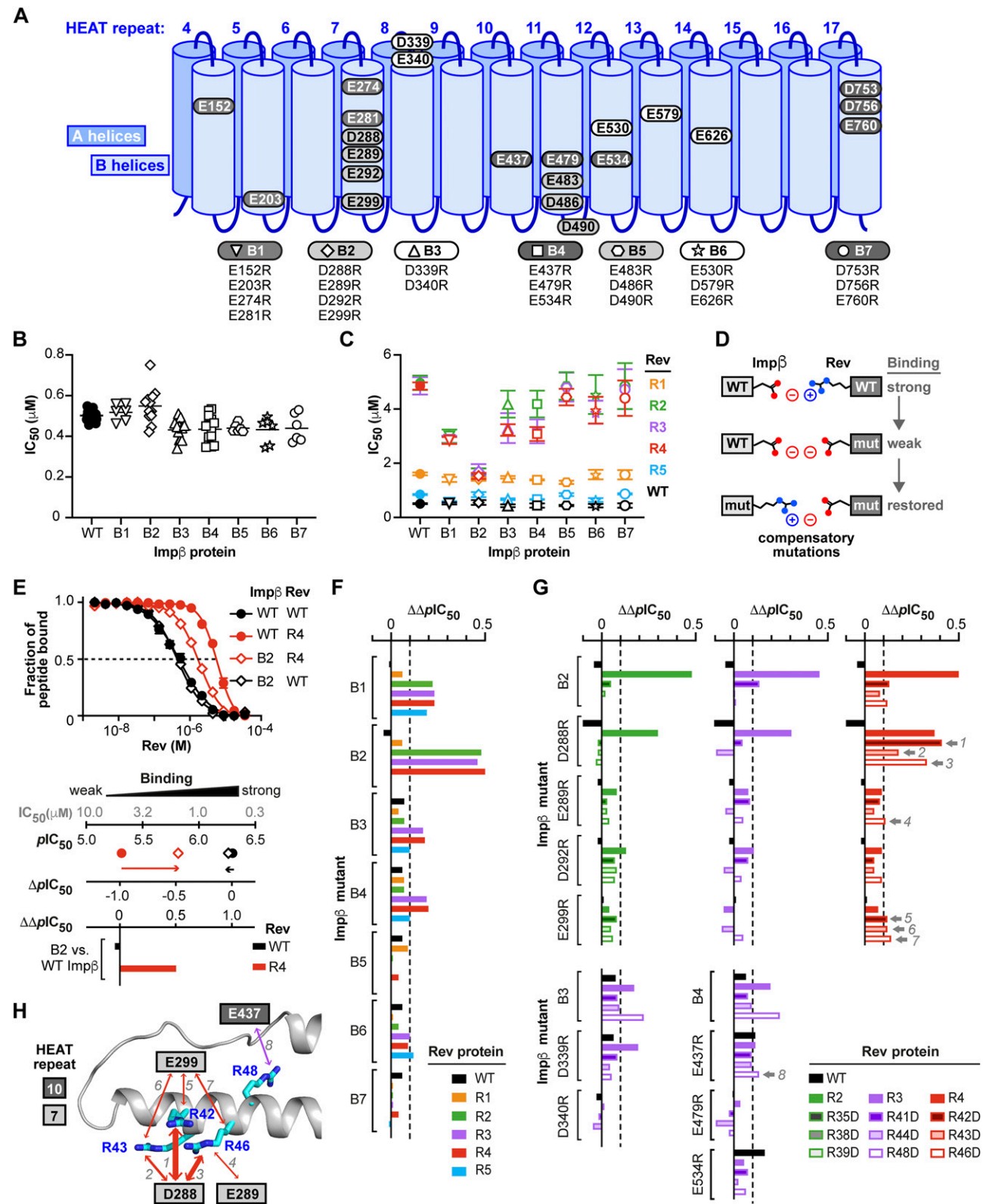

**Figure 10. Compensatory effects between charge-reversal mutants of Impβ and Rev.**
**(A)** Multiple point mutants of Impβ in which 2–4 Asp or Glu residues are replaced by Arg residues. **(B)** IC_{50} values obtained from FP inhibition assays measuring the ability of WT Rev to displace a Rev-NLS peptide from WT or mutant forms of Impβ. Results from individual assays are shown together with the mean. **(C)** The same assay performed with WT and charge-reversal mutant forms of Rev. Data for Rev mutants are colored as in Fig 8. Data shown are mean and SD values from 4 to 14 biological

contrast, all solutions from experiment 1 (with Rev in the N-site) placed these residues far apart and sterically inaccessible to each other (Fig S8A). Repeating these simulations with the Glu437$^{Imp\beta}$:Arg48$^{Rev}$ interaction explicitly included as an additional distance restraint yielded significantly worse docking results for Rev bound to the N-site (experiment 3) and considerably improved results for Rev bound to the C-site (experiment 4) (Figs S8C and D and S9C and D and Table S3). These findings suggest that the set of electrostatic interactions used as distance restraints in experiments 3 and 4 are more compatible with the C-site than with the N-site (Fig S8E).

To confirm this hypothesis, we performed exhaustive rigid-body sampling of Rev binding orientations on the surface of Imp$\beta$ within the N-site and C-site envelopes depicted in Fig 7F. The Imp$\beta$ structure was held fixed, whereas the Rev helical hairpin was rotated and translated in a full 6-dimensional search that maintained Lys20$^{Rev}$ within BS3 cross-linking distance of the group-1 lysines in HEAT repeats 1 and 2 (in the case of the N-site) or in repeat 19 (in the case of the C-site), yielding >3.5 and >2.8 million configurations, respectively. After excluding configurations with a severe steric overlap between Imp$\beta$ and Rev, the remaining configurations were used to measure C$\beta$-C$\beta$ distances between specific Imp$\beta$ and Rev residues. These measurements showed that the putative interactions illustrated in Fig 10H are individually compatible with Rev bound to either the N- or C-site because docking configurations were identified for both sites that brought the relevant charged residues within salt-bridge distance of each other (Fig S10A). However, no configurations were identified for the N-site in which interactions involving residue pairs Arg42$^{Rev}$:Asp288$^{Imp\beta}$ and Arg48$^{Rev}$:Glu437$^{Imp\beta}$ could simultaneously occur (i.e., configurations in which both C$\beta$-C$\beta$ distances were <14 Å), unlike the case for the C-site, where >80 such configurations were identified (Fig S10B). These findings confirm that the putative interactions depicted in Fig 10H have a much greater probability of being associated with the C-site than with the N-site.

In addition, the rigid-body sampling analysis also shed light on two of the BS3 cross-links (between Lys20$^{Rev}$ and Lys206$^{Imp\beta}$ or Lys537$^{Imp\beta}$) that we previously were unable to assign reliably to either the N- or C-site. The minimal C$\beta$-C$\beta$ distance measured between residues Lys20$^{Rev}$ and Lys206$^{Imp\beta}$ was 30.1 Å for all C-site configurations, compared with 12.5 Å for N-site configurations (Fig S10A), thereby assigning this cross-link with high probability to the N-site. Among the ~48,000 N-site configurations compatible with a Lys20$^{Rev}$:Lys206$^{Imp\beta}$ cross-link, only ~670 were compatible with a long cross-link between Lys20$^{Rev}$ and Lys537$^{Imp}$ (C$\beta$-C$\beta$ distance between 28.9 and 30 Å) and none with a shorter cross-link (C$\beta$-C$\beta$ distance <28.9 Å) (Fig S10C). In

contrast, the minimal C$\beta$-C$\beta$ distance for this residue pair was 13.7 Å for C-site configurations (Fig S10A), making it highly likely that the Lys20$^{Rev}$:Lys537$^{Imp\beta}$ cross-link is associated with the C-site.

## Structural model of Rev bound to the Imp$\beta$ C-site

We next filtered the results of our rigid-body docking experiment for configurations of Rev bound to the Imp$\beta$ C-site that were compatible with our BS3 cross-linking and compensatory mutagenesis data (Fig 11A). This resulted in a highly uniform set of parallel Rev configurations (Fig 11B and Table S4). To confirm this result, we ran HADDOCK using the BS3 cross-linking and compensatory mutagenesis data as interaction restraints to guide the docking procedure. Nearly all (97%) of the solutions obtained in the final refinement stage of docking yielded parallel configurations belonging to seven very similar clusters (Fig 11C and Table S5) that closely resembled the configurations obtained by rigid body docking (Fig 11D). As expected, both docking approaches yielded models that agreed well with our BS3 cross-linking data (Fig S11A) and placed residues related by compensatory charge-reversal effects in close proximity (Fig S11B). To represent the entire ensemble of configurations obtained by rigid-body sampling and HADDOCK, we calculated an "average" configuration that minimizes the change in orientation and position relative to each of the individual docking solutions obtained (Fig 11E). The resulting model closely resembles the top-ranked configurations obtained by rigid-body sampling and by HADDOCK (Fig S12). Comparing the individual docking configurations to the average model shows that the predicted location (i.e., translational coordinates of the centroid) of Rev on Imp$\beta$ (Fig 11E, lower inset) is relatively well defined, as are two of the rotational angles (yaw and pitch), whereas the angle about the long axis of Rev (roll angle) is somewhat more variable. A SAXS curve calculated from the average model agreed reasonably well ($\chi^2$ = 1.86 and 1.17 using Imp$\beta$ conformations 1UKL and 3W5K, respectively) with the SAXS data measured from the Imp$\beta$/Rev$^{OD}\Delta$ complex (Fig S13). Overall, these findings support a C-site Rev binding mode whereby the Rev helical hairpin is embraced by the C-terminal half of Imp$\beta$ with the long axis of Rev roughly parallel to the Imp$\beta$ superhelical axis and the Rev CTD positioned near the C-terminal end of the HEAT-repeat superhelix.

## Implications for Rev bound at the Imp$\beta$ N-site

We next wondered how the second Rev monomer might bind to Imp$\beta$. Because Rev is known to multimerize through homotypic

replicates (see also Table S2). **(D)** Compensatory effects between charge-reversal mutants. An interaction between an acidic residue on Imp$\beta$ and an Arg residue on Rev is disrupted by a charge-reversal (R→D) mutation of Rev. The interaction is restored by a charge-reversal (D/E→R) mutation on Imp$\beta$. **(E)** Derivation of $\Delta p$IC$_{50}$ values. *Top*: FP inhibition assays performed with WT Rev (black curves) or the R4 mutant (red curves) together with WT Imp$\beta$ (circles) or the B2 mutant (diamonds). *Bottom*: The ability of WT or mutant Rev to displace the Rev-NLS peptide from WT or mutant Imp$\beta$ is plotted as $p$IC$_{50}$ values and as the shift ($\Delta p$IC$_{50}$) relative to the value observed when WT Rev is assayed with WT Imp$\beta$. The $\Delta\Delta p$IC$_{50}$ value represents the shift in $\Delta p$IC$_{50}$ observed when the assay is performed with an Imp$\beta$ mutant instead of WT Imp$\beta$. A positive value of $\Delta\Delta p$IC$_{50}$ indicates that the tested Rev mutant more potently displaces the Rev-NLS peptide from the indicated Imp$\beta$ mutant than from WT Imp$\beta$. **(F, G)** Summary of $\Delta\Delta p$IC$_{50}$ values for the indicated combinations of Imp$\beta$ and Rev proteins. The dotted line at $\Delta\Delta p$IC$_{50}$ = 0.1 (corresponding to a 21% decrease in IC$_{50}$ between the mutant and WT Imp$\beta$) indicates the threshold used to identify compensatory mutations. Compensatory effects satisfying this criterion involving single-point mutants of Imp$\beta$ and Rev are marked by a gray arrow and numbered 1–8. For all eight of these mutant combinations, the mean shift in IC$_{50}$ value observed with the Imp$\beta$ single-point mutant relative to WT Imp$\beta$ was statistically significant according to a Dunnett's multiple comparisons test (P-values for mutant combinations 1–8 were <0.0001, 0.0002, 0.0001, 0.0022, 0.0368, 0.0084, <0.0001 and 0.0127, respectively). **(H)** Summary of compensatory interactions involving single Imp$\beta$ and Rev residues. **(G)** The numbered arrows correspond to the mutant combinations numbered 1–8 in (G). The thickness of arrows is proportional to the $\Delta\Delta p$IC$_{50}$ values of the interacting mutants.

## A Interactions used for Docking:

Rev Impβ Rev Impβ

K20 ↔ BS3 ↔ [ K537, K854, K857, K859, K867, K873 ]

R42 ⊕ ⊖ D288
R43 ⊕ ⊖ E299
R46 ⊕ ⊖ E289
R48 ⊕ ↔ ⊖ E437

**Rigid Body docking:**
- interactions used as constraints (strict distance cutoff applied)

**HADDOCK:**
- interactions used as restraints (contributing to total configuration energy)

## B Rigid-body docking

| Rank | |
|---|---|
| ■ | 1 |
| ■ | 2 |
| ■ | 3 |
| ■ | 4 |
| ■ | 5 |
| ■ | 6 |
| ■ | 7 |
| ■ | 8 |
| ■ | 9 |
| ■ | 10 |
| ■ | 11 |
| ■ | 12 |
| ■ | 13 |
| ■ | 14 |
| ■ | 15 |

## C HADDOCK

| Rank | |
|---|---|
| ■ | 1 |
| ■ | 2 |
| ■ | 3 |
| ■ | 4 |
| ■ | 5 |
| ■ | 6 |
| ■ | 7 |

## D Superposition

Rigid-body
HADDOCK

## E "Average" Configuration

CTD
±5.5 Å
±24°
±6.3 Å
±52°
±4.5 Å
±35°

pitch
yaw
roll

● Rigid-body
● HADDOCK

**Figure 11. Structural model of Rev bound to Impβ at the C-site.**
**(A)** BS3 cross-links and electrostatic interactions used as distance constraints for rigid body docking and as interaction restraints in program HADDOCK. **(B)** Configurations of the Impβ/Rev complex obtained by rigid body docking. The mean variation in Rev orientation and centroid position are 33° and <5 Å, respectively, relative to the top-ranked configuration (see also Table S4). **(C)** Configurations of the Impβ/Rev complex obtained using HADDOCK. For clarity, only 28 (the four lowest-energy structures in each of the seven clusters) of the 194 clustered solutions are shown. Solutions are colored according to the rank of the corresponding cluster. The mean variation in Rev orientation and centroid position are 34° and 4 Å, respectively (see also Table S5). **(D)** Superposition showing agreement between docking solutions obtained by rigid body sampling and by HADDOCK. **(E)** "Average" configuration of the Impβ/Rev complex that minimizes the deviation from models obtained by rigid-body and HADDOCK docking experiments. Compared with the average configuration, the Rev orientation in the individual rigid-body and HADDOCK configurations shows the greatest variation (by a maximum of 52°) in the angle about the long axis of Rev (roll angle) and smaller variations (up to 25–35°) about the two orthogonal angles (yaw and pitch). The approximate location of the Rev CTD is indicated. *Upper inset*: Principal axes and corresponding rotational angles of Rev. *Lower inset*: Centroids of Rev monomers obtained in rigid-body (green spheres) and HADDOCK (violet spheres) docking experiments. These centroids differ from that of the average model by a maximum of 5 or 6 Å along each of the principal axes of Rev.

"A–A" and "B–B" interactions, we structurally aligned both types of Rev homodimer with our average model of Rev bound to the Impβ C-site to see whether either dimer was compatible with the model (Fig S14A and B). Both alignments resulted in an implausible placement of the second Rev monomer: residue Lys20 on Rev helix α1 was very far from group-1 lysines in HEAT repeats 1 and 2, in disagreement with the BS3 cross-linking data; there was insufficient space to accommodate the Rev CTD; and, in the case of the Rev B–B homodimer, a severe steric clash occurred with Impβ (Fig S14B). These findings make it highly unlikely that the second Rev monomer binds Impβ by forming an A–A or B–B interface with the first bound Rev monomer. This conclusion is strongly supported by the ability of Impβ to bind two monomers of the Rev[OD] mutant (Figs 2D and E, 3B–D, and 4B), whose ability to form A–A and B–B dimers is compromised (Fig 2A).

For additional insights, we further exploited our rigid-body docking data. We structurally aligned the average model of Rev bound to the Impβ C-site with all Rev docking configurations that were compatible with BS3 cross-links involving the N-site (including the K20[Rev]-K206[Impβ] cross-link; ~48,000 configurations; Fig S10C) and retained those exhibiting no, or at most only a mild, steric overlap between the two Rev monomers (~18,000 configurations). As expected, the resulting Rev monomers all localize to the

N-terminal half of Impβ, with approximately one-third located within contact distance of the Rev monomer bound at the C-site (Fig S14C). The latter subset is characterized by two predominant configurations in which the two Rev monomers interact through their α1-α2 loops, with the second Rev monomer adopting either an antiparallel or a roughly perpendicular ("transverse") orientation (Fig S14D). Interestingly, in the latter case, the relative orientation of the two Rev monomers closely resembles the C–C homodimer arrangement recently described for Rev (67). Indeed, aligning the structure of a Rev C–C homodimer (PDB 5DHV) with our average model of Rev bound to the C-site yields only a minor steric clash with Impβ that can be relieved by a small rigid-body rotation of the dimer (Fig S14E), suggesting that Impβ could conceivably bind two Rev monomers sharing a C–C interface. A non-exhaustive examination of docking configurations revealed that several models of the 1:2 Impβ: Rev complex, including those with a C–C homodimer or an antiparallel arrangement for the two Rev monomers, are consistent with the SAXS data measured from the Impβ/Rev$^{OD}$Δ complex, supporting the plausibility of these models (Fig S15). However, additional data are required to establish the specific binding configuration of the second Rev monomer and whether it interacts with the first.

# Discussion

In this study, we investigated the interaction of human Impβ with HIV-1 Rev using diverse experimental and computational approaches. Gel shift, SEC-MALLS, and native MS experiments show that Impβ can bind either one or two Rev monomers, with ITC measurements reporting an ~10-fold difference in binding affinity between the two monomers. These findings are corroborated by cross-linking/MS data, which identify distinct Rev binding sites within the N- and C-terminal halves of Impβ. The ability of Impβ to engage two Rev monomers is in striking contrast with previously reported structures of Impβ/cargo complexes. These show Impβ loaded with only a single copy of its cargo (48, 58, 61, 63), except for SREBP-2, which binds Impβ as a preformed dimer (59). Unlike SREBP-2, the independent binding of Rev monomers revealed by gel shift and ITC analysis shows that Impβ does not associate with a preformed Rev dimer but binds one Rev molecule at a time, reminiscent of how the RRE binds Rev (65, 90).

Is the binding of Rev to the lower affinity site on Impβ merely an in vitro observation or does it also occur in virally infected cells? The concentration of Rev in the infected cell varies over the course of infection, initially starting low and gradually rising as viral transcription proceeds (14). The previously estimated concentration of cellular Impβ (1–2 μM) (44) and our measured $K_d$ values allow one to estimate how the fractional occupancy of Impβ's two Rev binding sites vary as a function of Rev concentration (Fig S16A) and hence to estimate the corresponding distributions of bound and unbound Impβ and Rev species (Fig S16B–D). These calculations predict that at the low Rev concentrations (<10 nM) expected during early phases of viral infection, the majority (75%) of Rev is bound to Impβ in a 1:1 complex, with ~10% of bound Rev localizing to the lower affinity site (Fig S16C and D). Both sites become increasingly occupied as the Rev concentration increases (Fig S16A), and at low

μM concentrations (below the critical level at which Rev multimerizes in vitro (103)), up to 38% of Impβ may be bound in a complex with two Rev monomers (Fig S16B). Taken together, these estimates suggest that the binding of Rev to Impβ's lower affinity site may occur to a significant extent in the virally infected cell.

Impβ is conformationally highly flexible, allowing it to adapt to and recognize diverse cargos through different regions of its HEAT repeat array. Although a C-terminal Impβ region (approximately repeats 7–19) recognizes the IBB domains of Impα (48) and Snurportin1 (61) and the DNA-binding domain of SREBP-2 (59), a middle region (repeats 5–14) recognizes Snail1, and an N-terminal region (approximately repeats 1–11) recognizes cyclin B1 (104), ribosomal protein L23 (105), and the parathyroid hormone–related protein PTHrP (58). The in vitro observation that cyclin B1 and the IBB domain of Impα can both bind to the same Impβ molecule demonstrates that Impβ can use distinct N- and C-terminal regions to bind two molecular cargos simultaneously (104). Our finding that Impβ contains two Rev binding sites involving opposite ends of the HEAT-repeat superhelix is consistent with this observation and agrees with a study reporting that Rev interacts with both an N-terminal (res. 1–396) and a C-terminal (res. 304–876) fragment of Impβ (44). Interestingly, the alternative Rev import receptor transportin-1 also contains two independent Rev binding sites (44). Conceivably, the ability of these import receptors to bind two Rev molecules might accelerate the nuclear import rate of Rev or increase its half-life in the cell (106), which could potentially modulate nuclear or cytosolic functions of Rev.

Molecular docking simulations informed by cross-linking/MS, SAXS and compensatory mutagenesis data allowed us to deduce a specific structural model for Rev bound at the C-site of Impβ. This model is admittedly based solely on in vitro data, and so it would be interesting to validate the results in vivo, for example, by examining the ability of Rev mutants with reduced Impβ-binding affinity to localize to the nucleus in a cell line in which Impβ is known to mediate Rev import. Our structural model of Rev bound at the C-site of Impβ overlaps sterically with that of RanGTP bound to the N-terminal half of Impβ (Fig S17A; presumably a Rev monomer bound at the N-site would also strongly overlap with RanGTP), accounting for the ability of RanGTP to compete with Rev for binding to Impβ (references 43 and 44 and Fig S3). Previous studies identified residues within the ARM of Rev (res. 35–50) as important for binding Impβ (42, 43, 44). Our mutagenesis and peptide-scanning experiments are consistent with these reports and further delineate the first half of the ARM (res. 35–41) as being particularly important for binding affinity. These residues mediate key interactions with stem IIB and with an adjacent site on the RRE (70, 74) as well as with stem IA (88), explaining why Impβ and the RRE bind Rev in a mutually exclusive fashion (42, 43, 107). Our compensatory mutagenesis experiments identified four acidic residues of Impβ (Asp288, Glu289, Glu299, and Glu437) as being potentially in direct contact with the Rev ARM. Subsets of these residues also mediate recognition of the Impα IBB domain (Asp288), SREBP-2 (Glu299, Glu437), and Snail1 (Asp288, Glu289, Glu437), suggesting that Rev exploits the same Impβ residues used to import cellular cargos.

Among the structurally characterized Impβ/cargo complexes, the predicted binding mode of Rev at the C-site most closely resembles that of the Impα IBB domain (Fig S17B). The long C-terminal helix of the IBB domain adopts a similar parallel orientation as that

predicted for Rev helix α2 but is shifted by 25 Å toward the C-terminal HEAT repeats, where it makes salt bridge interactions with the B helices of repeats 12–18. Besides its C-terminal helix, the IBB also contains an N-terminal extended moiety that makes extensive van der Waals and hydrogen-bonding interactions with the B helices of HEAT repeats 7–11. N-terminal truncations that remove one or both of the highly conserved Arg13 and Lys18 residues in this moiety severely abrogate binding to Impβ (108, 109). The Arg residues at the tip of Rev helix α2 in our structural model overlap closely with the N-terminal IBB moiety (Fig S17C), accounting for the ability of the IBB domain to compete with Rev for binding to Impβ (43) and raising the possibility that key basic residues in these two molecular cargos mediate similar interactions with Impβ.

Rev is known to multimerize through hydrophobic A–A and B–B interfaces (69, 73, 85, 110), which above a critical Rev concentration (~6 µM) mediate the in vitro assembly of helical filaments (84, 103, 111, 112). Our molecular docking simulations and the observation that Impβ binds two monomers of the WT and oligomerization-defective Rev$^{OD}$ mutant argue against the formation of an A–A or B–B interface between the two Impβ-bound Rev monomers. Interestingly, however, our data are compatible with the two bound Rev monomers sharing a weaker C–C interface, previously observed in Rev filaments and in crystallization contacts (67, 68). In the Rev-RRE assembly pathway, a C–C interaction between two RRE-bound Rev dimers has been proposed to mediate a "four-Rev" specificity checkpoint that precedes the association of additional Rev molecules (67, 113). This raises the speculative possibility that a C–C interface mediating recruitment of the second Rev monomer to Impβ might compete with the checkpoint C–C interaction and potentially regulate Rev-RRE assembly/disassembly. Alternatively, the two Rev monomers might constitute a previously unobserved interface that is specifically induced during Impβ-mediated nuclear import. A third possibility is that they do not share any Rev–Rev interface because our current binding data do not allow us to assert or exclude cooperativity between the two Rev binding sites with confidence.

In conclusion, our findings reveal that the interaction of Rev with Impβ is unlike that of cellular Impβ cargos in that it can interact through two independent binding sites that localize to the N- and C-terminal halves of the import receptor. Additional studies are needed to uncover the atomic details underlying Rev recognition by Impβ and the functional implications of the atypical binding stoichiometry.

## Materials and Methods

### Protein expression and purification

#### Impβ

N-terminally His-tagged Impβ was expressed from a pETM-11 based plasmid in *Escherichia coli* strain BL21 (DE3). Cells were grown in LB medium containing kanamycin (50 µg/ml) at 37°C until an OD$_{600}$ of 0.5–0.6, and protein expression was induced with 0.5 mM IPTG for 5 h at 30°C. Harvested cells were lysed by sonication at 4°C in lysis buffer (50 mM Hepes [pH 7.5], 150 mM NaCl, 10 mM imidazole, 10 mM MgCl$_2$, 2 mM β-mercaptoethanol [βME]) supplemented with

nuclease and protease inhibitors (10 µg/ml DNase I, 10 µg/ml RNase A, 2 mM PMSF, and 1 tablet/80 ml of complete EDTA-free protease inhibitor cocktail [Roche]). The soluble fraction was recovered by centrifugation (50,000g, 20 min, 4°C) and applied to a HisTrap FF NiNTA column (GE Healthcare). After washing with buffer A (50 mM Hepes [pH 7.5], 200 mM NaCl, 2 mM βME) containing 40 mM imidazole, proteins were eluted in the same buffer containing 300 mM imidazole. For untagged Impβ, the His-tag was then removed by overnight incubation with His-tagged TEV protease (1:20 w/w TEV/Impβ protein ratio) supplemented with 2 mM βME and 2 mM PMSF and recovered in the flow-through of a new NiNTA column. Both untagged and His-tagged Impβ were further purified using a Superdex 200 16/60 gel filtration column (GE Healthcare) in 50 mM Hepes (pH 7.5), 100 mM NaCl, 1 mM tris(2-carboxyethyl)phosphine (TCEP). Fractions containing pure Impβ were pooled, concentrated on an Amicon centrifugal filter (30 kDa cutoff; Millipore) and stored as aliquots at −80°C. The Impβ protein concentration was determined using a molar extinction coefficient of 79,051 M$^{-1}$ cm$^{-1}$, which was experimentally determined by quantitative amino acid analysis.

ICmpβ charge-reversal point mutations were generated by a PCR-based protocol adapted from the QuikChange site-directed mutagenesis method (Agilent Technologies). Impβ mutant proteins were expressed and purified as the WT and their molecular masses verified by LC/ESI mass spectrometry (Table S1).

### HIV-1 Rev

Rev proteins were expressed from a pET-28a based plasmid as an N-terminal His-GB1-TEV fusion as previously described (114). Protein expression was obtained from *E. coli* BL21 (DE3) cells grown in Auto-Induction Medium (115) containing kanamycin (50 µg/ml) for 12 h at 27°C. Harvested cells were lysed by sonication at 4°C, in lysis buffer (25 mM Hepes [pH 7.5], 200 mM NaCl, 100 mM Na$_2$SO$_4$, 10 mM MgCl$_2$, 10 mM imidazole, 0.1% Tween 20) containing nuclease and protease inhibitors (as described above for Impβ). The clarified lysate was incubated with RNAse T1 (20 U/µl; Roche), RNase A (20 µg/ml; Euromedex), and 2 M NaCl for 1.5 h at room temperature. After centrifugation (50,000g, 10 min, 4°C), the soluble fraction was applied to a HisTrap FF NiNTA column (GE Healthcare) and washed first with HSB B (2 M NaCl, 50 mM Tris [pH 8], 0.1% Tween 20, 2 mM βME, and 10 mM imidazole) followed by low-salt buffer B (including 250 mM NaCl and no Tween 20) containing 40 mM imidazole. Proteins were eluted in buffer B containing 300 mM imidazole and diluted with an equal volume of 50 mM Tris (pH 8) before loading onto HiTrap Heparin HP columns (GE Healthcare). After extensive washing with 50 mM Tris (pH 8) and 125 mM NaCl, elution was performed using 50% HSB (50 mM Tris [pH 8], 200 mM NaCl, 400 mM (NH4)$_2$SO$_4$, 100 mM Na$_2$SO$_4$, 2 mM βME) and 50% of 2 M NaCl. The His-GB1 tag was then removed from the Rev protein by an overnight incubation with His-tagged TEV protease (1:20 w/w TEV/Rev protein ratio) supplemented with 2 mM βME and 2 mM PMSF. The untagged Rev protein recovered in the flow-through after passing the sample through a new NiNTA column was then diluted with five volumes of 50 mM Tris (pH 8) and then concentrated by repeating the heparin chromatography step described above. Heparin elution fractions containing Rev were applied onto a Superdex 75 16/60 gel filtration column (GE Healthcare) in HSB containing 1 mM TCEP. Fractions containing pure Rev were pooled, concentrated on an Amicon

centrifugal filter (10 kDa cutoff; Millipore), and stored at −80°C. Rev protein concentration was determined using a molar extinction coefficient of 9476 $M^{-1}$ $cm^{-1}$, which was experimentally determined by quantitative amino acid analysis.

Rev[OD]Δ (Rev[4–69][V16D/I55N]) and Rev charge-reversal point mutations were generated by a PCR-based protocol adapted from the QuikChange site-directed mutagenesis method (Agilent Technologies). Rev mutant proteins were expressed and purified as the WT, and their molecular masses verified by LC/ESI mass spectrometry (Table S1).

### $^{15}N,^{13}C$-labeled Rev[OD] protein

*E. coli* BL21 (DE3) cells expressing Rev[OD] were grown in $M9/H_2O$ medium containing 10 g/l $Na_2HPO_4.7H_2O$, 3 g/l $KH_2PO_4$, 0.5 g/l NaCl, 1 g/l $^{15}NH_4Cl$, 2 g/l $^{13}C$ D-glucose, 1 mM $MgSO_4$, 0.1 mM $CaCl_2$, 0.01 mM $MnCl_2$, 0.05 mM $ZnSO_4$, 1× MEM vitamin solution (Thermo Fisher Scientific), and kanamycin (50 $\mu g/ml$). Precultures were started from a glycerol stock in LB + kanamycin, agitated in a 100 mL Erlenmeyer flask at 180 rpm, 37°C. After 8 h, cells were collected for 15 min at 1,650$g$ and inoculated in $M9/^{15}N/^{13}C$ medium and grown overnight at 37°C. The following day, precultures were inoculated in $M9/^{15}N/^{13}C$ medium (1:35) + kanamycin and grown at 37°C until reaching an $OD_{600}$ of 0.7. Protein induction was obtained by adding 1 mM IPTG and growing the culture overnight at 16°C. Cells were harvested by centrifugation and resuspended in denaturing lysis buffer (8 M urea, 50 mM $NaH_2PO_4/Na_2HPO_4$ [pH 7.4], 500 mM NaCl, 5 mM $\beta$-mercaptoethanol, 0.02% $NaN_3$, 25 mM imidazole) supplemented with 1 mM PMSF and one tablet of complete EDTA free (Roche). After sonication at 4°C, lysed cells were centrifuged at 50,000$g$, 4°C for 30 min. The supernatant containing $^{15}N,^{13}C$-labeled His-GFP-TEV-Rev[OD] denatured protein was loaded on a HisTrap FF nickel column. After washing with denaturing lysis buffer at 5 ml/min, the flow rate was decreased to 1 ml/min and Rev refolding obtained by applying a linear gradient from 100% lysis buffer to 100% refolding buffer (50 mM $NaH_2PO_4/Na_2HPO_4$ [pH 7.4], 500 mM NaCl, 5 mM $\beta$-mercaptoethanol, 0.02% $NaN_3$). After an additional wash of refolding buffer, refolded protein was eluted with elution buffer (50 mM $NaH_2PO_4/Na_2HPO_4$ [pH 7.4], 500 mM NaCl, 5 mM $\beta$-mercaptoethanol, 0.02% azide, 500 mM imidazole). The His-GFP-TEV fusion was then removed from the Rev[OD] protein using His-tagged TEV protease (1:20 w/w TEV:Rev protein ratio), by overnight incubation at 4°C in dialysis buffer (50 mM $NaH_2PO_4/Na_2HPO_4$ [pH 7.4], 150 mM KCl, 1 mM DTT, 1 mM EDTA, 0.02% $NaN_3$) using 10 kDa cutoff dialysis membrane. The untagged $^{15}N,^{13}C$-labeled Rev[OD] protein was recovered in the flow-through after passing the sample through a new HisTrap FF column, centrifuged (50,000$g$, 30 min, 4°C), and concentrated by ultrafiltration (Amicon, 10 kDa cutoff) before further purifying on a Superdex S75 16/60 gel filtration column (GE Healthcare) in 25 mM $NaH_2PO_4/Na_2HPO_4$ (pH 7.4), 150 mM KCl, 1 mM DTT, 1 mM EDTA, 0.02% $NaN_3$. The Rev[OD] protein was dissolved in 90% $H_2O/10\%$ $D_2O$ with gel filtration buffer, and the protein concentration was adjusted to 90 $\mu M$ in 150 $\mu l$ in the absence or presence of Imp$\beta$ (at equimolar ratio).

### TEV protease

N-terminally His-tagged TEV protease bearing the point mutation S219V was expressed and purified as previously described (114).

### SEC/MALLS

SEC was performed on a high-pressure liquid chromatography system equipped with an LC-20AD pump, an autosampler SIL20-ACHT storing the samples at 4°C, a communication interface CBM-20A (Shimadzu), using a Superdex 75 10/300 GL column (GE Healthcare Bio-Sciences AB), thermostated at 20°C in an oven XL-Therm (WynSep), and equilibrated with running buffer (for Rev proteins: 200 mM NaCl, 400 mM $(NH_4)_2SO_4$, 100 mM $Na_2SO_4$, 50 mM Tris [pH 8], and 1 mM Tris(2-carboxyethyl)phosphine hydrochloride [TCEP]; for Imp$\beta$/Rev complexes: 50 mM Hepes [pH 7.5], 200 mM NaCl). Volumes of 50 $\mu l$ of each sample (60–710 $\mu M$) were injected at a flow rate of 0.5 ml/min. The elution profiles were followed on-line at 280 nm (SPD-M20A Shimadzu) by the refractive index and static and dynamic light scattering using a laser emitting at 658 nm (Optilab rEX, miniDAWN TREOS, and Dynapro Nanostar detectors, respectively, Wyatt Technology). Data were analyzed using ASTRA V 5.3.4.20 software (Wyatt Technology). The basic equation for light scattering, restricted to diluted solutions of small particles (<20 nm), reduces to I = KcM × (∂n/∂c)2, where I is the excess intensity of scattered light, K is an optical parameter, c is the weight concentration, and ∂n/∂c is the refractive index increment. The concentration was evaluated on-line from the refractive index signal, combined, for two-component analysis, with the 280 nm absorbance. From the amino acid composition and using Sedfit (https://sedfitsedphat.github.io/sedfit.htm), we obtained ∂n/∂c of 0.184 ml $g^{-1}$ and extinction coefficients of 0.647 liters $g^{-1}$ $cm^{-1}$ for Rev variants; ∂n/∂c of 0.186 ml $g^{-1}$ and extinction coefficients of 0. 817 liters $g^{-1}$ $cm^{-1}$ for Imp$\beta$; and ∂n/∂c of 0.186 ml $g^{-1}$ and extinction coefficients of 0. 797 liters $g^{-1}$ $cm^{-1}$ for Imp$\beta$/Rev complexes.

### DSF

HIV-1 consensus B Rev (15-mer) peptides were obtained from the NIH AIDS Research and Reference Reagent Program (Cat. no. 6445). Peptides were dissolved at 5 mM in water or 1–80% DMSO, as recommended. DSF experiments were performed on a CFX96 touch real-time PCR detection system (Bio-Rad) using 0.2-ml 8-tube PCR strips (#TBS0201) and optical flat 8-cap strips (#TCS0803). Assay samples were done over a temperature range starting from 20°C up to 100°C, in increments of 0.5°C, with 15 $\mu M$ Imp$\beta$ mixed with 75 $\mu M$ Rev-peptide and 1× SYPRO Orange (Molecular Probes) diluted in $H_2O$. The melting temperature ($T_m$) values of Imp$\beta$ in the presence of each Rev-peptide were determined from the fluorescence first derivative of the melting curves using CFX Maestro software. The wavelengths used for SYPRO Orange excitation and emission were 470 and 570 nm, respectively.

### Native gel shift assay

Imp$\beta$ and Rev proteins were initially diluted in PBS to 25 $\mu M$ and incubated in a 1:1 to 1:5 Imp$\beta$:Rev ratio (3.7 $\mu M$ Imp$\beta$, 10 $\mu l$ final volume) for 5 min. After adding 1 $\mu l$ of native loading buffer (62.8 mM Tris HCl [pH 6.8], 40% glycerol, 0.01% bromophenol blue), 2.5 $\mu l$ of samples were analyzed on a 10% TGX (Bio-Rad) gel, run under native conditions (0.5X TBE buffer, 4°C, 100 V, 120 min). Gels were

stained with Coomassie blue and scanned on a ChemiDoc MP gel imaging system (Bio-Rad).

## ITC

Calorimetric experiments were performed on a MicroCal iTC200 calorimeter (GE Healthcare) at 20°C while stirring at 800 rpm. All proteins were buffer exchanged by dialysis into 50 mM Hepes (pH 7.5), 200 mM NaCl. Typically, 35–50 $\mu$M Rev$^{OD}$ or Rev$^{OD}\Delta$ and 100–130 $\mu$M Imp$\beta$ proteins were placed in the cell and syringe, respectively. Titrations consisted of 26 identical injections of 1.5 $\mu$l made at time intervals of 3 min. ITC data were analyzed with NITPIC, SEDPHAT, and GUSSI public-domain software packages [116] using a single- or two-binding site models. The first data point was excluded from the analysis. Experiments were done in triplicate, and the variability was estimated to be less than 5% in the binding enthalpy and 10% in both the binding affinity and the number of sites.

## Glutaraldehyde cross-linking

A total of 5 $\mu$M Imp$\beta$ was mixed with 0–20 $\mu$M Rev in 50 mM Hepes (pH 7.5), 100 mM NaCl. Subsequently, 0.004% (vol/vol) of glutaraldehyde (ref. G7776; Sigma-Aldrich) was added and the reaction was incubated for 5 min at RT. The cross-linking reaction was stopped by adding 90 mM Tris (pH 8), and the mixture analyzed by native and denaturing gel electrophoresis and mass spectrometry.

## Mass spectrometry

### Liquid chromatography/electrospray ionization mass spectrometry (LC/ESI-MS)

LC/ESI-MS was performed on a 6210 LC-TOF spectrometer coupled to an HPLC system (Agilent Technologies). All solvents used were HPLC grade (Chromasolv; Sigma-Aldrich). TFA was from Acros Organics (puriss., p.a.). Solvent A was 0.03% TFA in water; solvent B was 95% acetonitrile-5% water-0.03% TFA. Immediately before analysis, protein samples were diluted to a final concentration of 5 $\mu$M with solvent A and then desalted on a reverse-phase C8 cartridge (Zorbax 300SB-C8, 5 $\mu$m, 300 $\mu$m ID 5 mm; Agilent Technologies) at a flow rate of 50 $\mu$l/min for 3 min with 100% solvent A and subsequently eluted with 70% solvent B for MS detection. MS acquisition was carried out in the positive ion mode in the 300–3,200 $m/z$ range. MS spectra were acquired and the data processed with MassHunter workstation software (v. B.07.00, Agilent Technologies) and with GPMAW software (v. 7.00b2, Lighthouse Data).

### Matrix-assisted laser desorption ionization/time-of-flight mass spectrometry (MALDI-TOF MS)

MALDI-TOF mass spectra were measured with an Autoflex mass spectrometer (Bruker Daltonics) operated on linear positive ion mode. External mass calibration of the instrument, for the $m/z$ range of interest, was carried out using the monomeric (66.4 kDa) and dimeric (132.8 kDa) molecular ions of BSA (reference 7030; Sigma-Aldrich) as calibrants. Before MS analysis the protein samples (concentration around 10 $\mu$M; before and after cross-linking) were submitted to buffer exchange against 20 mM Tris (pH 7.5), 50 mM NaCl, using Vivaspin devices (Sartorius, cut-off of

30 kDa for Imp$\beta$ alone and 10 kDa for Imp$\beta$/Rev). The buffer-exchanged protein samples were then mixed in a 1:2, 1:5 or 1:10 (vol/vol) ratio with sinapinic acid matrix (Sigma-Aldrich; 10 mg/ml in water/acetonitrile/TFA, 50/50/0.1, vol/vol/v) and 1–2 $\mu$l were deposited on the target and allowed to air dry at RT. Mass spectra data were processed with flexAnalysis software (v.3.0; Bruker Daltonics).

### Native mass spectrometry

Imp$\beta$ and Rev proteins were individually buffer exchanged by gel filtration using a Superdex 200 5/150 (Imp$\beta$) or Superdex 75 5/150 (Rev) column pre-equilibrated with either 250 mM (Imp$\beta$), 400 mM (Rev$^{OD}$ and Rev$^{OD}\Delta$), or 800 mM (RevWT) ammonium acetate. The individual proteins or mixtures of Imp$\beta$ with Rev were incubated at 20°C for 15 min before MS analysis. Protein ions were generated using a nanoflow electrospray (nano-ESI) source. Nanoflow platinum-coated borosilicate electrospray capillaries were bought from Thermo Electron SAS. MS analyses were carried out on a quadrupole time-of-flight mass spectrometer (Q-TOF Ultima; Waters Corporation) modified for the detection of high masses [117, 118, 119]. The following instrumental parameters were used: capillary voltage = 1.2–1.3 kV, cone potential = 40 V, RF lens-1 potential = 40 V, RF lens-2 potential = 1 V, aperture-1 potential = 0 V, collision energy = 30–140 V, and microchannel plate (MCP) = 1,900 V. All mass spectra were calibrated externally using a solution of cesium iodide (6 mg/ml in 50% isopropanol) and processed with the MassLynx 4.0 software (Waters Corporation) and Massign software package [120].

### LC-MS/MS analysis and BS3 cross-link identification

A total of 4.1 $\mu$M of His-tagged Imp$\beta$ was mixed with 8.2 $\mu$M of WT Rev in 50 mM Hepes (pH 7.5), 150 mM NaCl before adding 25 $\mu$M of bis(sulfosuccinimidyl)suberate (BS3) (ref. 21580; Thermo Fisher Scientific). The reaction was incubated for 30 min at room temperature and stopped with 25 mM NH$_4$HCO$_3$. The reaction mixture was then subjected to SDS–PAGE, and the band corresponding to the cross-linked complex was excised from the Coomassie-stained gel. Disulfide bridges were reduced using dithiothreitol (Roche). Free thiol groups were subsequently alkylated with iodoacetamide. Proteins were digested with trypsin (Promega) at 37°C overnight. Reactions were stopped with formic acid (FA), and peptides were extracted from the gel with 5% formic acid (FA) in 50% acetonitrile. Acetonitrile content was reduced in a vacuum centrifuge before LC-MS/MS analysis. Peptides were analyzed on an UltiMate 3000 HPLC RSLC nano system coupled to a Q Exactive HF-X quadrupole-orbitrap mass spectrometer, equipped with a Nanospray Flex ion source (all Thermo Fisher Scientific). Peptides were trapped on a PepMap C18 cartridge (5 mm × 300 $\mu$m ID, 5 $\mu$m particles, 100 Å pore size) and separated on a PepMap C18 column (500 mm × 75 $\mu$m ID, 2 $\mu$m, both Thermo Fisher Scientific) applying a linear gradient from 2% to 35% solvent B (80% acetonitrile, 0.08% FA; solvent A 0.1% FA) at a flow rate of 230 nl/min over 120 min. The mass spectrometer was operated in data-dependent mode: survey scans were obtained in a mass range of 350–1,600 m/z with lock mass on, at a resolution of 60,000 at 200 m/z, and an AGC target value of 3E6. The 10 most intense ions were collected subsequently with an isolation

width of 1.6 Th for maximal 250 ms and fragmented at 28% normalized collision energy. Spectra were recorded at a target value of 2E4 and a resolution of 60,000. Peptides with a charge of +1,+2, or >+7 were excluded from fragmentation, the peptide match feature was set to preferred, the exclude isotope feature was enabled, and selected precursors were dynamically excluded from repeated sampling for 20 s within a mass tolerance of 5 ppm. Raw data were searched with pLink 2.3.5 (121) against the sequences of the 16 most abundant protein hits from the MaxQuant search (version 1.6.0.16) (122), as well as the sequences of target proteins and common contaminant sequences. BS3 was selected as the cross-linking chemistry. Carbamidomethyl on Cys was set as fixed; oxidation of Met and protein N-terminal acetylation as variable modifications. Enzyme specificity was selected according to the protease used for digestion. Search results were filtered for 1% FDR on the PSM level, limiting the precursor mass deviation to 5 ppm. To remove low-quality–peptide spectrum matches, an additional e-Value cutoff of < 0.001 was applied.

### NMR

Assignment was obtained using $^{15}N,^{13}C$-labeled samples (90 $\mu M$) using BEST-TROSY three-dimensional experiments HNCO, intra-residue HN(CA)CO, HN(CO)CA, intra-residue HNCA, HN(COCA)CB, and intra-residue HN(CA)CB (123) recorded on a Bruker spectrometer equipped with a cryoprobe operating at 20°C and a $^1H$ frequency of 600 and 700 MHz. Spectra were processed using NMRPipe (124) and analysed in Sparky (Goddard, T and Kneller, D SPARKY 3. University of California, San Francisco). MARS (125) was used for spin system identification, combined with manual verification.

### SEC-SAXS

Size-exclusion chromatography coupled with small-angle X-ray scattering (SEC-SAXS) experiments were conducted on the BM29 beamline at the European Synchrotron Radiation Facility using a Superdex 200 Increase 5/150 GL column. A total of 30 $\mu l$ of protein sample (containing 25 $\mu M$ of Imp$\beta$ with or without 125 $\mu M$ of Rev), previously spun (21,000$g$, 10 min), were injected and the flow rate maintained at 0.35 ml/min using 50 mM Tris buffer (pH 8), 100 mM NaCl. Acquisitions (1 s/frame) were recorded during 10 and 1,000 s for buffer and protein, respectively. Individual frames were processed using the software BsxCUBE, yielding individual radially averaged curves of normalized intensity versus scattering angle. Time frames corresponding to each elution peak were combined to give the average scattering curve for each measurement. The averaged spectrum of the buffer was subtracted from the averaged spectrum of protein before data analysis. Primary data reduction was performed, and model-independent parameters $R_g$ and $D_{max}$ were determined using the program PRIMUS (126). The program CRYSOL (127) was used to calculate the theoretical scattering from individual crystal structures.

### Fluorescence polarization (FP) assays

Fluorescence polarization assays were performed as previously described (114) in black 384-well plates (Greiner ref. 781076; Greiner Bio-One) on a CLARIOstar plate reader (BMG Labtech) using a 540 ± 20 nm excitation filter, a 590 ± 20 nm emission filter, and an LP565

dichroic mirror. The inhibition assays were carried out in a total volume of 10 $\mu l$, with 195 nM Imp$\beta$, 200 nM Rev-NLS-TAMRA peptide, and 4.3 nM–35 $\mu M$ Rev variants. Curves obtained with each Imp$\beta$ and Rev protein were independently fitted, and the IC$_{50}$ values were determined with GraphPad Prism 9 software using equation $F = F_{min} + (F_{max} - F_{min})/(1 + 10^{\wedge}((LogIC_{50}-X) \times HillSlope))$, where $F_{min}$ and $F_{max}$ are the fluorescence polarization values at saturating Rev concentration and in the absence of Rev, respectively, and $X$ is the common logarithm of the Rev concentration.

### Molecular docking with HADDOCK

The helical hairpin domain of Rev (PDB 2X7L chain M) was docked onto Imp$\beta$ (PDB 1UKL chain A) using the HADDOCK2.2 webserver (102). HADDOCK uses biochemical information on interacting residues as ambiguous interaction restraints (AIRs) to drive the docking. Residues directly implicated or potentially involved in mediating the interaction are defined as "active" and "passive," respectively. Each AIR is defined between an active residue of one molecule and specific active or passive residues of the other. Passive residues for Rev were defined as all residues (res. 9–65) except those designated as active, whereas passive residues for Imp$\beta$ were defined as all solvent-accessible residues on the concave inner surface of the protein. (See footnote 2 of Table S3 for the explicit list of residues). The set of AIRs and corresponding cutoff distances for each docking experiment are listed in Tables S3A and S5A. For BS3 cross-linking restraints, the cutoff distance between the C$\beta$ atoms of cross-linked lysine residues was set at 30 Å. For AIRs between Rev Arg residues and Imp$\beta$ Asp or Glu residues that yielded compensatory mutagenesis effects, the C$\beta$-C$\beta$ cutoff distance was set to a value between 5 and 8 Å, depending on the associated $\Delta\Delta pIC_{50}$ value (Fig 10F), with shorter distances assigned to residue pairs yielding larger $\Delta\Delta pIC_{50}$ values. The HADDOCK docking protocol consists of randomization of orientations and rigid body energy minimization, semirigid simulated annealing in torsion angle space, and final refinement in the Cartesian space with explicit solvent. For each docking experiment, a total of 1,000 complex configurations were calculated in the randomization stage. The best 200 solutions were then used in the semirigid and final refinement stages, in which the side-chain and backbone atoms of Imp$\beta$ and Rev were allowed to move. The final structures were ranked according to HADDOCK score and clustered by pairwise backbone RMSD. Clusters were then ranked according to the average of the four best-scoring structures within each cluster. In all docking experiments, the top-ranked cluster scored substantially better than the next-ranked cluster and was also the largest in size.

### Rigid body docking

Rigid body sampling was performed with the Rev helical hairpin domain (PDB 2X7L chain M) and Imp$\beta$ (PDB 1UKL chain A) using in-house Unix shell and Fortran scripts combined with the program Lsqkab from the CCP4 suite (128). Two independent docking jobs were run for Rev bound either to the N-site or the C-site of Imp$\beta$. For each job, the Imp$\beta$ structure was held fixed, whereas Rev was rotated and translated in a six-dimensional search. The translational search was performed on a 4 Å grid spacing over grid points located within 35 Å of C$\alpha$ atoms of the Imp$\beta$ lysine residues that

formed BS3 cross-links with Rev Lys20 (group-1 residues in Fig 7B; Lys residues 23, 62, and 68 in the case of the N-site and Lys residues 854, 857, 859, 867, and 873 in the case of the C-site). In total, 879 and 714 grid points were used for the N-site and C-site docking jobs, respectively. Rotations of Rev were performed with the C$\beta$ atom of Rev Lys20 held fixed at the origin of the rotation, which was centered on each grid point. The rotational search employed a 15° polar angle step size (4,056 orientations per grid point), resulting in >3.5 and >2.8 million configurations for the N- and C-sites, respectively.

For each docking configuration, the steric overlap between Rev and Imp$\beta$ was estimated using a mask overlap procedure, as follows. A mask for Imp$\beta$ was generated at 5 Å sampling resolution by (i) rounding the coordinates of each atom to the nearest multiple of 5 Å, (ii) sorting the resulting file on the rounded coordinates (i.e., sorting for increasing value of $x$ then $y$ then $z$), and (iii) removing duplicate coordinates from the sorted file. These same steps were then used to generate a mask for Rev for each docking configuration. For each configuration a combined mask covering both the Imp$\beta$ and Rev structures was generated by concatenating the individual Imp$\beta$ and Rev mask files, sorting the resulting file on the mask coordinates, and removing duplicate coordinates from the sorted file. The overlap between the Imp$\beta$ and Rev masks was then given by $N_{overlap} = N_{Imp\beta} + N_{Rev} - N_{comb}$, where $N_{Imp\beta}$, $N_{Rev}$ and $N_{comb}$ are the number of mask points in the Imp$\beta$, Rev and combined masks, respectively. $N_{overlap}$ values above 15 were found to correspond to a large steric overlap between Imp$\beta$ and Rev that could not be relieved by simply reorienting the side chains of these two proteins. Docking configurations yielding an $N_{overlap} > 15$ were thus rejected from further analysis. The number of configurations that survived this steric overlap criterion was ~1.5 and 1.9 million for the N- and C-site, respectively. For each surviving configuration, C$\beta$-C$\beta$ distances were calculated for the 10 Imp$\beta$/Rev residue pairs indicated in Fig S10A and used to generate the boxplots and distance distribution diagrams shown in Fig S10A and B.

For the docking of Rev to the C-site of Imp$\beta$, configurations were subsequently filtered according to the distance cutoff criteria indicated in Table S4A, resulting in the 15 configurations shown in Fig 11B. These configurations were then ranked as follows. For each of the eight Imp$\beta$/Rev residue pairs mediating compensatory mutation effects (left column of Table S4A), the minimal distance ($D_{min}$) between the Imp$\beta$ and Rev residues was calculated over all sterically allowed side-chain rotamers. (In the few cases in which the two closest Imp$\beta$ and Rev rotamers sterically overlapped, the value of $D_{min}$ was set to 2.3 Å). Each $D_{min}$ value was then weighted by the corresponding $\Delta\Delta pIC_{50}$ value (Fig 10G), and the weighted mean ($<\Delta\Delta pIC_{50}*D_{min}>$) was calculated over all eight residue pairs. Docking configurations were then ranked in increasing value of $<\Delta\Delta pIC_{50}*D_{min}>$, which ranged from 4.6 Å for the top-ranked solution to 7.31 Å for the solution ranked 15[th].

### Average docking model of Rev bound to the C-site of Imp$\beta$

The average configuration of Rev bound to the Imp$\beta$ C-site was calculated from a total of 43 individual docking configurations. These consist of the 15 configurations obtained by rigid body docking (indicated in Fig 11B and Table S4) and 28 configurations (the four lowest energy structures from each of the seven clusters) from the final HADDOCK experiment (Fig 11C and Table S5). The 43 configurations were superimposed onto a reference Imp$\beta$ structure, and the centroid of all 43 Rev structures was calculated. The 43 Rev structures were then translated so as to have their individual centroids coincident at this common centroid position. A Rev search model (PDB 2X7L chain M) was then centered on this position in an arbitrary orientation and a rotational search was performed using a 10° polar angle step size (12,996 orientations). For each orientation, a distance score was calculated between representative atoms located along the three principal axes of the Rev search model and the corresponding atoms of the 43 individual Rev structures. The orientation yielding the lowest score was selected as the average docking configuration.

## Supplementary Information

## Acknowledgements

We acknowledge the European Synchrotron Radiation Facility for provision of synchrotron radiation facilities. We thank ME Brennich and M Tully for assistance in using beamline BM29; L Shamseddine for help with protein purification; A Le Roy and C Ebel for SEC/MALLS and AUC analyses; JP Andrieu for amino acid analysis; H Dawi for initial HADDOCK experiments; A Sinz, C Ihling and M Schäfer for exploratory XL-MS experiments; and T Gossenreiter, D Anrather, and M Hartl from the Mass Spectrometry Facility at Max Perutz Labs for BS3 cross-linking analyses, which were performed using the Vienna BioCenter Core Facilities instrument pool. This study was supported by grants (to C Petosa) from the ANRS and Fondation de France and PhD funding (to D Spittler) from the Agence Nationale de Recherches sur le Sida et les Hépatites Virales (ANRS) and the Commissariat à l'Énergie Atomique et aux Énergies Alternatives (CEA). This work used the platforms of the Grenoble Instruct-ERIC center (ISBG; UAR 3518 CNRS-CEA-UGA-EMBL) within the Grenoble Partnership for Structural Biology (PSB), supported by FRISBI (ANR-10-INBS-0005-02) and GRAL, financed within the University Grenoble Alpes graduate school (Ecoles Universitaires de Recherche) CBH-EUR-GS (ANR-17-EURE-0003).

### Author Contributions

D Spittler: investigation.
R-L Indorato: investigation.
E Boeri Erba: formal analysis, investigation, and writing—review and editing.
E Delaforge: investigation.
L Signor: investigation.
SJ Harris: investigation.
I Garcia-Saez: investigation.
A Palencia: formal analysis, investigation, and writing—review and editing.
F Gabel: formal analysis, validation, and writing—review and editing.
M Blackledge: supervision, validation, and writing—review and editing.
M Noirclerc-Savoye: formal analysis, supervision, investigation, and writing—original draft, review, and editing.
C Petosa: conceptualization, formal analysis, supervision, funding acquisition, investigation, and writing—original draft, review, and editing.

## Conflict of Interest Statement

The authors declare that they have no conflict of interest.

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
