## [Reviewer comments · Life Science Alliance]

Life Science Alliance

Binding stoichiometry and structural model of the HIV-1 Rev/Importin beta complex

Didier Spittler, Rose-Laure Indorato, Elisabetta Boeri Erba, Elise Delaforge, Luca Signor, Simon Harris, Isabel Garcia Saez, Andres PALENCIA, Frank Gabel, Martin Blackledge, Marjolaine Noirclerc-Savoie, and Carlo Petosa

DOI: <https://doi.org/10.26508/lsa.2022.01431>

Corresponding author(s): Carlo Petosa, Institut de Biologie Structurale and Marjolaine Noirclerc-Savoie, Institut de Biologie Structurale

Review Timeline:

Submission Date:	2022-02-28
Editorial Decision:	2022-03-02
Revision Received:	2022-06-27
Editorial Decision:	2022-07-19
Revision Received:	2022-07-21
Accepted:	2022-07-22

Transaction Report:

Please note that the manuscript was reviewed at Review Commons and these reports were taken into account in the decision-making process at Life Science Alliance.

March 2, 2022

Re: Life Science Alliance manuscript #LSA-2022-01431

Carlo Petosa
Institut de Biologie Structurale
71 Avenue des Martyrs
Grenoble 38000
France

Dear Dr. Petosa,

Thank you for submitting your manuscript entitled "Binding stoichiometry and structural model of the HIV-1 Rev/Importin beta complex" to Life Science Alliance. We invite you to re-submit the manuscript, revised according to your Revision Plan.

Thank you for this interesting contribution to Life Science Alliance. We are looking forward to receiving your revised manuscript.

Sincerely,

B. MANUSCRIPT ORGANIZATION AND FORMATTING:

Point-by-point response

We are grateful to the two Reviewers for their comments and suggestions for improving the manuscript. The text below uses the following convention: *comments by the Reviewers are in blue italics*, our responses are in black and text quoted from the revised manuscript is in brown.

Reviewer #1 (Evidence, reproducibility and clarity):

Spittler et al report results from a series of in vitro studies designed to characterize the interaction of the HIV-1 Rev protein with Importin-beta. Using recombinant proteins and a series of experimental approaches including size exclusion chromatography, native and crosslinker mass spectrometry, isothermal titration calorimetry, BEST-TROSY NMR spectroscopy, small angle X-ray scattering, extensive mutational analysis, differential scanning fluorimetry and fluorescence polarization they determine that 2 Rev molecules interact with importin-beta via two independent sites. Molecular docking simulation with constraints from the biophysical analysis generates a structural model that suggests that the interaction of Rev with the C-terminal portion of importin-beta resembles that of importin-alpha's IBB domain. This agrees with previous reports that the IBB domain can compete with Rev for importin-beta binding, which is nice support for a prediction coming from their model. The specific binding configuration of the second Rev monomer to the N-terminal portion of importin-beta could not be explicitly modelled, but is obvious incompatible with the A-A or B-B dimer forms of Rev.

Overall, this is a convincing, well-written and highly detailed analysis of Rev interaction with importin-beta. The authors have done a great deal of high quality work here, and generated new information about these interactions. The figures appear well rendered, and the data seem convincing.

Major comments: None

Minor comments:

1) The limitation of the work is that it is all performed in vitro, with recombinant proteins. It would be interesting to compare the ability of the Rev mutants to localize to the nucleus in vivo and compare with the ability to bind importin-beta. I think the authors have performed more than enough work here already, and suggest that this can be left for a future study. Nevertheless, I encourage adding a small statement in the discussion about weaknesses (all experiments done in vitro) and possible lines of study in vivo to support the conclusions.

As suggested by the reviewer, we have modified the Discussion to include the underlined statement below:

Molecular docking simulations informed by crosslinking/MS and compensatory mutagenesis data allowed us to deduce a specific structural model for Rev bound at the C-site of Imp β . This model is admittedly based solely on in vitro data, and so it would be interesting to validate the results in vivo, for example, by examining the ability of Rev mutants with reduced Imp β -binding affinity to localize to the nucleus in a cell line in which Imp β is known to mediate Rev import.

2) Although the authors mention that their model is likely compatible with the C-C dimer of Rev, I don't think that they provide any evidence of co-operative binding by the second Rev monomer. To my mind, this makes it seem less likely that the Rev monomers interact with each other in any significant manner. For example, my take is that binding of the Rev oligomerization deficient mutants to the second site is comparable to wt. This may also deserve some brief commentary in the discussion.

The C-C interface of Rev is a relatively weak, variable interface that has only been observed as a secondary interface between subunits of a Rev complex that already share an A-A or B-B interface; hence, any cooperativity due to C-C interactions between two Imp β -bound Rev monomers may be difficult to detect. More generally, whereas positive cooperativity between two binding sites with similar ligand binding affinities is relatively easy to show, cooperativity between two binding sites having different binding affinities (as is the case for the two

Rev-binding sites on Imp β) is much more difficult to demonstrate. Based on our available binding data, we cannot reliably state whether the two Rev monomers bind in a cooperative or non-cooperative manner.

As suggested by the referee, we briefly comment on this point by modifying the Discussion to include the underlined statement below:

This raises the speculative possibility that a C-C interface mediating recruitment of the second Rev monomer to Imp β might compete with the checkpoint C-C interaction and potentially regulate Rev-RRE assembly/disassembly. Alternatively, the two Rev monomers might constitute a previously unobserved interface that is specifically induced during Imp β -mediated nuclear import. A third possibility is that they do not share any Rev-Rev interface, since our current binding data do not allow us to assert or to exclude cooperativity between the two Rev binding sites with confidence.

Reviewer #1 (Significance (Required)):

Significance: this data provides an incremental advance in our detailed understanding of the interaction of HIV-1 Rev with importin-beta, but does yet not provide a final definitive structural model of this interaction. The interaction between Rev and importin-beta was originally described and studied in the 1990s yet the detailed molecular basis of this interaction is not fully understood, so there is some novelty here. This interaction is critical for Rev function, which is critical for the HIV replicative cycle. Theoretically, the Rev/importin-beta interaction could be blocked by inhibitors as a means of controlling HIV infection (<https://doi.org/10.1016/j.bbamcr.2010.07.010>) and the information from this paper could move rationale design of such inhibitors a little closer to reality.

Audience: This data will be of interest to at least a subset of those working in the HIV field and likely those interested in nuclear import.

Our expertise: nuclear import, virology, biochemistry, molecular biology

We are grateful that Reviewer #1 appreciates the value of our study. While a high-resolution structure of the Imp β /Rev complex has so far proved elusive, our structural model nevertheless captures essential features of this complex that enable previously reported functional data to be rationalized in structural terms.

Reviewer #2 (Evidence, reproducibility and clarity (Required)):

The HIV-1 Rev (Regulator of Expression of the Virion), essential for viral replication, mediates the nuclear export of unspliced viral RNA. Hence, Rev protein carrying nuclear localization signal (NLS) needs to be imported into the nucleus and carries out its nuclear export function. The NLS of Rev has been identified, and multiple nuclear import factors, including transportins and Importin- β , have been shown to mediate the nuclear import of Rev in distinct cell types. In this study, Spittler et al. applied multiple biochemical biophysical approaches to study the molecular basis of Rev and Importin- β interaction that mainly occurs in T lymphoma cells.

Previous studies have shown that Rev has a tendency to form a higher-order assembly. Hence, in this manuscript, the authors used oligomerization-deficient Rev mutants to study the interaction with the Importin- β .

Rev mutants did form complexes with Importin- β as they co-migrated in the size exclusion chromatography. Moreover, results of native MS, native gel, and ITC demonstrated a 1:2 molar ratio of Importin- β to Rev, suggesting two Rev binding sites on Importin- β . Analysis of NMR chemical shift perturbations identified an ARM motif on Importin- β , serving as the major Rev binding site. Additionally, XL-MS combined with mutagenesis analyses showed that Rev binds to the putative minor binding site in the C-terminus of Importin- β via the charge-charge interaction. However, the binding of the Rev does not substantially change the Importin- β conformation revealed by the SAXS analysis. Overall, by combining the biochemical and biophysical results, a simulated docking model of Importin- β and Rev is presented, suggesting an atypical binding model.

To avoid confusion, we note that, contrary to the third last sentence in the above paragraph, our data identify the binding site within the C-terminal half of Imp β as the putative major, not minor, Rev binding site.

MAJOR CONCERNS:

1. Authors only applied LC/ESI-MS to determine the stoichiometry of WT Rev-Importin- β complex (1:1 or 1:2) and the rest of the studies was conducted using oligomerization-deficient Rev. However, it is important to have the biochemically reconstituted WT Rev-Importin- β complex as a reference and analyze it by size exclusion chromatography (in solution) as WT Rev protein tends to oligomerize in solution. Next, other methods (e.g. analytical ultracentrifugation) addition to LC/ESI-MS need to be applied to validate stoichiometry of "WT" Rev-Importin- β complex. Without having a stable WT complex, it is really not convincing that Rev and Importin- β form a 1:1 or 1:2 protein complex. Studying how Rev mutant that cannot oligomerize interacts with Importin- β may not reflect the physiological conditions.

Reviewer 2 is concerned that our study provides insufficient evidence that Imp β can bind two Rev monomers because the experiments informing on stoichiometry were primarily performed using Rev^{OD}, which contains two point mutations (V16D and I55N), rather than WT Rev. As requested by the reviewer, we have addressed this concern by including three additional experiments in the revised manuscript, and by performing a fourth (not included in the revision), as detailed below.

(1a) The SEC experiment with WT Rev requested by Reviewer 2 has been added to the manuscript as **Fig 2C** and compared with the initial SEC experiment with Rev^{OD} in **Fig 2B**. As expected, the control experiments with unbound Rev^{OD} and Rev^{WT} (magenta curves) yielded different chromatograms because Rev^{WT} precipitates under the buffer conditions used for this experiment. (This also explains why the peak height of the Imp β /Rev^{WT} complex is lower than that of the Imp β /Rev^{OD} complex). Consequently, we included an additional control showing the elution profile of Rev^{WT} in a high-salt buffer (HSB, purple curve). Although Rev^{WT} elutes earlier than Rev^{OD} because it forms soluble multimers in HSB (in agreement with our SEC/MALLS data shown in **Fig 2A**), the shift in elution volume between free Rev^{WT} and Imp β -bound Rev^{WT} is clear (compare top and bottom gel images in **Fig 2C**). Importantly, the elution volumes of the Imp β /Rev^{WT} and Imp β /Rev^{OD} complexes are essentially identical (compare green curves in **Fig 2B** and **C**), suggesting that these two complexes have the same stoichiometry.

Figure 2B,C. (B) SEC analysis of an Imp β /Rev^{OD} complex. *Top*, elution profiles of samples containing Imp β (blue), Rev^{OD} (magenta) or a mixture of Imp β and Rev^{OD} (green). Fractions collected are indicated in brown. Chromatography was performed using a Superdex 200 5/150 Increase GL column. *Bottom*, SDS-PAGE analysis of the indicated fractions. (C) SEC analysis of an Imp β /Rev^{WT} complex. Elution profiles are shown for Imp β (blue), Rev^{WT} (magenta) and a mixture of Imp β and Rev^{WT} (green). Since most free

Rev^{WT} forms insoluble aggregates in the buffer conditions used and hence is not detected in the elution, the elution profile of Rev^{WT} analysed on a column pre-equilibrated in high-salt buffer (HSB) was included as an additional reference (purple).

(1b) Our original manuscript included gel shift data showing that Imp β can bind either one (Complex 1) or two (Complex 2) monomers of the Rev^{OD} mutant. We performed the corresponding experiment using WT Rev. The new data are included in **Fig 2D** of our revised manuscript. They confirm that Imp β can bind either one or two monomers of WT Rev:

Figure 2D. Native gel analysis showing the association of Imp β with one or more monomers of Rev^{WT} (top) and Rev^{OD} (bottom).

(1c) In addition we have performed SEC/MALLS analysis of the Imp β /Rev^{WT} complex and included the results in the revised manuscript as **Fig 2E**. Analysis of Imp β in the absence or presence of an excess of WT Rev reveal elution peaks associated with molar masses which are consistent (considering the experimental accuracy of the MALLS technique) with the ability of Imp β to either bind one or two monomers of Rev, depending on the amount of Rev added:

Figure 2E. SEC/MALLS analysis of Imp β in the presence and absence of Rev. Elution curves recorded at 280 nm and molar mass distributions determined by MALLS are shown for His-tagged Imp β in the absence (blue) and presence of either a 2-fold (light green) or 4-fold (dark green) excess of Rev^{WT}. The observed masses of 124 and 137 kDa are consistent with those expected for 1:1 (116.2 kDa) and 1:2 (129.4) Imp β :Rev stoichiometry. Chromatography was performed using a Superdex 200 10/30 GL column. Injected concentrations were 20 μ M Imp β and 40 or 80 μ M Rev^{WT}.

(1d) Reviewer 2 mentioned analytical ultracentrifugation (AUC) as a possible method to validate the stoichiometry of the Imp β /Rev^{WT} complex. We have performed these experiments and show the results below. Not surprisingly, unbound Rev forms oligomers and larger aggregates that yield increasingly complex sedimentation profiles as the protein concentration is increased (**Fig A** below). This makes it difficult to interpret

the sedimentation profiles observed for samples containing a mixture of Imp β and Rev and hence to deduce the distributions and stoichiometries of the resulting complexes (**Fig B**). While the overall data are not incompatible with the presence of a 1:2 Imp β :Rev complex among the components in the mixture, we feel that the results are not sufficiently conclusive to bring any added value to our study and so we have not included them in the revised manuscript.

Figure showing AUC analyses of Imp β , Rev^{WT} and the Imp β /Rev^{WT} complex (not included in manuscript).

Sedimentation velocity experiments were performed on a Beckman XL-I analytical ultracentrifuge equipped with an AN-50 TI rotor (Beckman Instruments) at 20°C, using 100 or 400 μ L samples loaded into 3 and 12 mm path-length centerpieces with Sapphire windows (Nanolitics), respectively, and centrifuged at 42,000 rpm. All samples were prepared in a buffer containing 44 mM HEPES pH 7.5, 6 mM TRIS pH 8, 194 mM NaCl, 52 mM NH₄SO₄, and 13 mM Na₂SO₄. Interferences were measured in continuous scan mode during sedimentation, and data were processed with Redate software (v. 1.0.1) and analyzed in terms of $c(s)$ distributions using SEDFIT (v. 14.1) and Gussi (1.2.1). Theoretical $s_{20,w}$ values were calculated from PDB files using Hullrad software (Fleming PJ, Fleming KG. Biophys J. 2018, 114:856-869). Experimental $c(s)$ values were determined using partial specific volumes of 0.739, 0.727 and 0.737 ml/g for Imp β , Rev and the Imp β /Rev complex, respectively, a solvent density of 1.016 g/ml and a viscosity of 1.058 cp. **(A)** Sedimentation velocity analysis of unbound Imp β (10 μ M) and unbound Rev tested at 3, 10 and 20 μ M concentration. Unbound Imp β primarily sedimented as a monomer with an $s_{20,w}$ of 4.9, comparable to the theoretical value ($s_{20,w}$ = 5.0) calculated from the atomic structure (taken from PDB 3W5K), with a minor fraction sedimenting as a dimer ($s_{20,w}$ = 7.6). Unbound Rev formed a mixture of oligomers and larger aggregates at all concentrations tested. *Inset*, magnified view of the boundary fractions between $s_{20,w}$ values of 3.5 and 7.5. **(B)** Sedimentation velocity analysis of Imp β (10 μ M) in the absence and presence of varying concentrations of Rev (from 3 to 60 μ M). Theoretical $s_{20,w}$ values for the 1:1 and 1:2 Imp β :Rev complexes indicated in the inset were calculated from the structural models shown in **Fig S13B** and **Fig S15B** (top panel) with the Rev C-terminal domain modeled in a random conformation calculated using the Bax group server <https://spin.niddk.nih.gov/bax/nmrserver/pdbutil/ext.html>. These values predict that the $s_{20,w}$ value should decrease slightly (from 5.0 to 4.9) when one Rev monomer is bound and increase (to 5.4) when two monomers are bound. While the observed data show a clear increase in the $s_{20,w}$ value as the Rev concentration is increased (thus arguing against a simple 1:1 interaction model), the data defy reliable interpretation in terms of a mixture of simple stoichiometric species, presumably because of the complex multimerization behaviour of Rev.

(1e) We additionally wish to clarify that the LC/ESI-MS data referred to in Reviewer 2's comment were actually native MS data, not LC/ESI-MS data. Native MS represents the state-of-the-art for determining the accurate mass and stoichiometry of macromolecular complexes (See, e.g., Ref. 91 and refs therein). The fact that native MS experiments identify Imp β species bound to either 1 or 2 monomers of WT Rev constitutes compelling evidence for the existence of these complexes. Finally, as mentioned in the Discussion, our findings that Imp β can bind two Rev molecules and that the binding sites localize to N- and C-terminal regions of Imp β are consistent with a previous study reporting that Rev is able to bind both an N-terminal and C-terminal fragment of Imp β (Ref.44). These observations combined with the additional new data included in our revised manuscript provide decisive evidence that Imp β binds two Rev molecules.

2. That Importin- β contains two Rev binding sites revealed by the ITC studies is not promising (Fig. 3). Authors proposed that Importin- β has two Rev binding sites simply because the fitting of two binding site model is better than a model consisting of one binding site (a better K_d value?). The biphasic nature of the isotherm indicates two binding sites, but the biphasic profiles are not obvious in Fig. 3. Additionally, what is the physiological relevance of $\sim 10 \mu\text{M}$ K_d value for the second binding site (Rev^{OD})? Since WT Rev can self-oligomerize, how can it bind to the second binding site of Importin- β with such a low affinity?

Reviewer 2 raises several concerns about the ITC data and derived K_D values, which we address below as three separate points.

2a. That Importin- β contains two Rev binding sites revealed by the ITC studies is not promising (Fig. 3). Authors proposed that Importin- β has two Rev binding sites simply because the fitting of two binding site model is better than a model consisting of one binding site (a better K_d value?).

The original version of the manuscript admittedly stated this conclusion without showing the poorly fitting binding model. We have rectified this by including an additional figure, **Fig S5**, that compares the fitting of a binding isotherm using models comprising either one or two binding sites. Clearly, the curve on the left fits the data poorly, whereas that on the right gives a convincing fit. Thus, the data are inconsistent with a single-binding site model but are adequately explained by a model comprising two binding sites.

Figure S5. ITC data are consistent with two classes of Rev binding sites on Imp β .

Representative ITC profile for the binding of Imp β to Rev^{OD} showing that a model consisting of two non-symmetric classes of binding sites (right) yields a better fit to the normalized binding isotherm than a model consisting of a single class of binding site (left), as highlighted by the data points circled in red.

2b. The biphasic nature of the isotherm indicates two binding sites, but the biphasic profiles are not obvious in Fig.3.

Whereas a biphasic isotherm is consistent with the presence of two or more binding sites, the presence of two binding sites does not necessarily guarantee a biphasic isotherm. This was illustrated in a 2009 study [Freire et al., *Methods Enzymol.* 455:127-155 (2009) PMID 19289205], in which isotherms were simulated for a macromolecule (M) containing two ligand (L) binding sites. The shape of these isotherms depends on the overall association constants, β_1 and β_2 , for the binding reactions $M + L \rightarrow ML$ and $M + 2L \rightarrow ML_2$, whose values are given by $\beta_1 = [ML]/([M][L])$ and $\beta_2 = [ML_2]/([M][L]^2)$, respectively. Four representative scenarios for the values of β_1 and β_2 were considered (See Figure below). Case A corresponds to two independent binding sites with identical ligand binding affinities; cases B and D correspond to either two independent sites that differ in binding affinity or two identical sites with negative cooperativity; and, case C corresponds to two identical sites with positive cooperativity. As seen below, only case D yielded a biphasic isotherm, while the other three cases yielded uniphase isotherms. The situation in our study is analogous to case B.

Figure 5.2 from Freire et al., 2009. Simulated titrations for a macromolecule with two binding sites.

Similarly, in a more recent study [Brautigam. *Methods* 76:124-136 (2015). PMID 25484338] isotherms were simulated for a macromolecule with two ligand binding sites ($M + 2L \rightarrow ML + L \rightarrow ML_2$) using a binding model with two nonsymmetric sites and no cooperativity, which is analogous to the situation in our study. The isotherm obtained for the case in which the first binding site had a four-fold higher affinity ($K_{D1} = 100$ nM) than the second ($K_{D2} = 400$ nM), is shown below and is clearly uniphase, not biphasic.

Figure 6A from Brautigam, 2015. Simulated uniphase two-site ITC data.

The above examples confirm that a uniphasic isotherm is compatible with the presence of two ligand binding sites. Hence, the ability of Imp β to bind two Rev monomers is entirely consistent with the isotherm shown in **Figure 4** of our manuscript.

2c. Additionally, what is the physiological relevance of $\sim 10 \mu\text{M}$ K_d value for the second binding site (RevOD)? Since WT Rev can self-oligomerize, how can it bind to the second binding site of Importin- β with such a low affinity?

We thank the reviewer for this question, which motivated us to consider the amount of Rev predicted to associate with the two binding sites on Imp β inside the cell. Our findings are included in the revised manuscript as a new figure, **Fig S16**, shown below, and described in an additional paragraph in the Discussion.

To address the question we estimated the concentration of bound and unbound Imp β and Rev species as a function of Rev concentration given the K_d values experimentally determined by ITC. The intracellular concentration of Imp β , [Imp β]_{cell}, has been estimated to be 1-2 μM (ref. 44), and so we chose an intermediate value of 1.5 μM for our calculations. Although a literature survey did not reveal any estimates for the Rev concentration in virally infected cells, this is known to vary over the course of infection, initially starting low and gradually rising as viral transcription proceeds (e.g., ref. 14). For any given total Rev concentration, the concentration bound to each site on Imp β at equilibrium can be readily calculated from the measured K_d values. Dividing these concentrations by [Imp β]_{cell} yields the fractional occupancy of each binding site, shown in **Fig S16A**. The critical concentration (c_{crit}) above which Rev self-oligomerizes to form filaments *in vitro* has been reported to be 6 μM (ref. 100). **Fig S16A** shows that at subcritical Rev concentrations a significant fraction of sites 1 and 2 on Imp β can be occupied by Rev (up to 87% and 43%, respectively). Similarly, **Fig S16B**, which summarizes the distribution of bound and unbound Imp β species, shows that a significant fraction (up to 38%) of Imp β can be bound simultaneously to two Rev monomers at concentrations where Rev does not self-oligomerize (black curve). Thus, the fact that WT Rev can self-oligomerize does not prevent Rev from significantly occupying either or both binding sites on Imp β .

The issue of physiological relevance is more appropriately addressed by considering the predicted distributions of the various bound and unbound Rev species (**Fig S16C and D**). **Fig S16C** shows that at low (nano- and sub-nanomolar) Rev concentration, most of Rev is bound to Imp β (black curve), with 66% and 8% of all Rev molecules bound at sites 1 and 2, respectively (blue and green curves). These fractions remain constant for all concentrations below approximately 0.2 and 1 μM , respectively. As the total Rev concentration rises above these levels Imp β becomes progressively saturated with Rev and the fraction of unbound Rev increases correspondingly (gray curve). **Fig S16D** shows the fraction of total bound Rev present in either a 1:1 or a 1:2 Imp β /Rev complex. At low Rev concentrations (<100 nM), the fractions of bound Rev that occupy either site 1 or site 2 in a 1:1 complex reach an asymptotic plateau at 90% and 10%, respectively; i.e., approximately one in ten Rev molecules recognized by Imp β is bound at site 2. As the Rev concentration rises above 100 nM an increasingly significant fraction of bound Rev is located within a 1:2 Imp β :Rev complex, and this fraction can reach 58% before the concentration at which Rev aggregates is attained (black curve).

In summary, these analyses reveal that at the low Rev concentrations that likely characterize early stages of viral infection, a significant fraction (8%) of Rev is predicted to be bound at site 2. At the higher Rev concentrations that may characterize later stages of infection, 1:2 Imp β :Rev complexes can account for over half of all bound Rev and over one-third of the total cellular Imp β (**Fig S16B and D**) before the critical Rev concentration is reached at which filament formation occurs. Thus, despite the relatively large K_d value for site 2, the elevated intracellular concentration of Imp β ensures that Rev binding at site 2 is significant in the physiological context.

In addition to **Fig S16**, our revised manuscript now includes the following additional paragraph in the Discussion that summarizes the above findings:

Is the binding of Rev to the lower affinity site on Imp β merely an *in vitro* observation or does it also occur in virally infected cells? The concentration of Rev in the infected cell varies over the course of infection, initially starting low and gradually rising as viral transcription proceeds [14]. The previously estimated concentration of cellular Imp β (1-2 μM) [44]

and our measured K_d values allow one to estimate how the fractional occupancy of Imp β 's two Rev-binding sites vary as a function of Rev concentration (Fig S16A), and hence to estimate the corresponding distributions of bound and unbound Imp β and Rev species (Fig S16B-D). These calculations predict that at the low Rev concentrations (<10 nM) expected during early phases of viral infection, the majority (75%) of Rev is bound to Imp β in a 1:1 complex, with approximately 10% of bound Rev localizing to the lower affinity site (Fig S16C and D). Both sites become increasingly occupied as the Rev concentration increases (Fig S16A), and at low μ M concentrations (below the critical level at which Rev multimerizes *in vitro* [100]) up to 38% of Imp β may be bound in a complex with two Rev monomers (Fig S16B). Taken together, these estimates suggest that the binding of Rev to Imp β 's lower affinity site may occur to a significant extent in the virally infected cell.

Figure S16. Estimated concentrations of Imp β and Rev species as a function of total Rev concentration.
(A) Fractional occupancy of Rev-binding sites 1 and 2 on Imp β . The intracellular concentration of Imp β , $[\text{Imp}\beta]_{\text{cell}}$, has been estimated at 1-2 μM [44] and is indicated by the black dashed line at 1.5 μM . The critical concentration,

c_{crit} , above which Rev self-oligomerizes to form filaments *in vitro* has been reported to be 6 μM [100] and is indicated by a red dashed line. The concentration of Rev in virally infected cells varies over the course of infection, initially starting low and gradually rising as viral transcription proceeds [14]. Because of the large value of $[\text{Imp}\beta]_{cell}$ only a small fraction of site 1 or site 2 is bound when Rev is in the low nM range, but this fraction rises quickly as Rev enters the high nM - low μM range. At subcritical concentrations Rev can occupy up to 87% and 43% of sites 1 and 2, respectively. **(B)** Fraction of $\text{Imp}\beta$ that is unbound (gray), in complex with one Rev monomer bound at either site 1 (blue) or site 2 (green) or in complex with two Rev monomers (black). A significant fraction (up to 38%) of $\text{Imp}\beta$ can be bound simultaneously to two Rev monomers at subcritical Rev concentrations. **(C)** Fraction of Rev that is unbound (gray), bound to $\text{Imp}\beta$ at site 1 (blue) or site 2 (green), or bound at either site (black). At low (nano- and sub-micromolar) Rev concentration, most of Rev (75%) is bound to $\text{Imp}\beta$, with 66% and 8% of all Rev molecules bound at sites 1 and 2, respectively. These fractions remain constant for concentrations below approximately 0.2 and 1 μM , respectively. As the total Rev concentration rises above these levels $\text{Imp}\beta$ becomes progressively saturated with Rev and the fraction of unbound Rev increases correspondingly. **(D)** Fraction of total $\text{Imp}\beta$ -bound Rev that is bound in a 1:1 complex at site 1 (blue), site 2 (green) or either site (white circles) or bound in a 1:2 complex (black). At Rev concentrations below ~ 100 nM, the fractions of bound Rev that occupy either site 1 or site 2 in a 1:1 complex are approximately 90% and 10%, respectively. As the Rev concentration rises above 100 nM an increasingly significant fraction of bound Rev is located within an $\text{Imp}\beta$: Rev_2 complex, and this fraction can reach 57% before c_{crit} is attained. For panels **(A-D)** the concentrations of bound and unbound $\text{Imp}\beta$ and Rev species were calculated using the equations $K_{d1} = (r-x-y)(b-x)/x$ and $K_{d2} = (r-x-y)(b-y)/y$, where r is the total Rev concentration, x and y are the Rev concentrations bound to $\text{Imp}\beta$ sites 1 and 2, respectively, b is $[\text{Imp}\beta]_{cell}$ (taken as 1.5 μM), and K_{d1} (0.61 μM) and K_{d2} (5.3 μM) are the K_d values determined by ITC for sites 1 and 2, respectively (**Fig 4A**). The fractional occupancy (probability of the bound state), p_1 and p_2 , of each binding site shown in **(A)** was given by $p_1=x/b$ and $p_2=y/b$. The fractions of $\text{Imp}\beta$ species represented by the gray, blue, green and black curves in **(B)** were calculated as the joint probabilities $P_{none}=(1-p_1)(1-p_2)$, $P_{site1}=p_1(1-p_2)$, $P_{site2}=(1-p_1)p_2$ and $P_{both}=p_1p_2$, respectively. The fractions of total Rev represented by the gray, blue, green and black curves in **(C)** were calculated as $(r-x-y)/r$, x/r , y/r and $(x+y)/r$, respectively. The fractions of total Rev bound in an $\text{Imp}\beta$ /Rev complex represented by the gray, blue, green and black curves in **(D)** were calculated as $(P_{site1} + P_{site2})/D$, P_{site1}/D , P_{site2}/D and $2P_{both}/D$, respectively, where $D = P_{site1} + P_{site2} + 2P_{both}$.

3. The cross-linking mass spec analysis revealed residues in the N and C-termini of Importin- β are involved in Rev binding. Based on the distance constraint of cross-linkers, the authors therefore suggested two binding site of Rev on Importin- β . However, how WT Rev (full length) folds and arranges in the complex are unknown. One still cannot exclude that Importin- β is in a configuration that wraps around a single Rev protein and thereby residues in the N and C termini are involved in the interaction to one Rev protein.

We certainly agree that residues in the N- and C-termini of $\text{Imp}\beta$ may interact with the same Rev monomer. Indeed, our structural model of Rev bound to the C-site of $\text{Imp}\beta$ shows precisely this: the top of the Rev hairpin domain localizes close to $\text{Imp}\beta$'s N-terminal HEAT repeats, while the bottom localizes close to its C-terminal repeats (**Fig 11E**). Nevertheless, the XL-MS experiment provides compelling evidence that two molecules of Rev interact with $\text{Imp}\beta$. These data show that a single Rev lysine residue, Lys20, forms crosslinks with lysines located near the N-terminus (including residues K23 and K62) and C-terminus (including residues K867 and K873) of $\text{Imp}\beta$. These two sets of lysines are so widely separated on the surface of $\text{Imp}\beta$ that it is **physically impossible** for a single Rev Lys20 residue to be within crosslinking distance of both groups. This is true irrespective of how other Rev residues are spatially distributed; i.e., how the Rev protein folds and arranges in the complex is irrelevant.

In the relatively extended known $\text{Imp}\beta$ conformations that are consistent with our SAXS data, the $\text{C}\alpha$ atoms of N- and C-terminal $\text{Imp}\beta$ lysine residues that crosslink with Rev K20 are separated by a straight-line (Euclidean) distance of 75-87 \AA and by a solvent-accessible surface distance (SASD) of 84-103 \AA (See **Fig S6B**, top right and bottom right panels). These distances are more than twice the generally accepted upper limit for the BS3 distance constraint. Even in the most compact known conformation of $\text{Imp}\beta$ (chain C in PDB entry 3LWW; see left panel

of **Fig S6A**) the SASD between these two groups of residues is between 70 and 84 Å (see **Fig S6B**, top left panel), signifying that they cannot both be within crosslinking distance of the same lysine residue.

In summary, the fact that two widely separated regions of Imp β form crosslinks to the same lysine residue on Rev strongly supports the existence of two distinct Rev binding sites on Imp β .

4a. The final model proposed that two monomeric Rev interacts with N (weak affinity) and C (strong affinity)-termini of Importin- β . Since Rev has a tenancy to oligomerize, how does it "dissociate" to the monomeric protein and interacts with Importin- β , particularly the N-terminal binding site with a low binding affinity?

See the response to point 2c above. Our analyses show that, because of the high Imp β concentration in the cell, a significant fraction of Rev is predicted to be bound at the weaker binding site even at very low Rev concentrations far below the critical threshold at which it self-oligomerizes.

4b. Additionally, is the Importin- β -mutant Rev complex dissociated in the presence of RanGTPase? Authors can conduct this experiment as well as superimpose their model with Importin- β -cargo complex structures to corroborate whether Importin- β uses a canonical way/or an atypical binding mode to interact with Rev.

We have performed additional gel shift experiments that show that both Rev monomers are released from Imp β upon incubation with RanGTP. The data are included in the revised manuscript as **Fig S3**. Identical results were obtained with both the mutant and WT forms of Rev (**Fig S3A** and **B**, respectively). The Results section have also been modified to include the following text:

Gel shift experiments (performed with either Rev^{OD} or Rev^{WT}) showed that RanGTP disrupted the formation of both the 1:1 and 1:2 complexes, indicating that RanGTP competes with both Rev monomers for binding to Imp β (**Fig S3**).

Figure S3. RanGTP competes with both monomers of Rev for binding to Imp β .

Samples containing a mixture of Imp β , RanGTP and either **(A)** Rev^{OD} or **(B)** Rev^{WT} were analysed by native gel electrophoresis (top) or SDS-PAGE (bottom) followed by Coomassie blue staining. Protein concentrations used were 2.5 μ M for Imp β , 5 μ M for RanGTP and either 2.5 or 5 μ M for Rev proteins, indicated by (+) and (++) , respectively. Compared to unbound Imp β (lanes 1, 11 and 18), incubating Imp β with Rev resulted in shifted and supershifted bands corresponding to 1:1 and 1:2 Imp β /Rev complexes, marked by light and dark green arrowheads, respectively (lanes 4, 7, 9, 14 and 16). Incubating Imp β with RanGTP yielded a band that migrated with intermediate mobility (purple arrowhead, lane 5). When all three Imp β , RanGTP and Rev proteins were co-incubated, the bands corresponding to the Imp β /Rev complexes were absent or only faintly detected whereas strong intensity was observed for the Imp β /RanGTP band, showing that RanGTP prevents Imp β from binding either monomer of Rev.

4c. Authors can [...] superimpose their model with Importin- β -cargo complex structures to corroborate whether Importin- β uses a canonical way/or an atypical binding mode to interact with Rev.

The requested superimposition of atomic structures was already presented in the initial manuscript (**Fig S17**) and described in the Discussion:

Among the structurally characterized Imp β /cargo complexes, the predicted binding mode of Rev at the C-site most closely resembles that of the Imp α IBB domain (**Fig S17B**). The long C-terminal helix of the IBB domain adopts a similar parallel orientation as that predicted for Rev helix α_2 but is shifted by 25 Å towards the C-terminal HEAT repeats, where it makes salt bridge interactions with the B helices of repeats 12-18. Besides its C-terminal helix, the IBB also contains an N-terminal extended moiety that makes extensive van der Waals and hydrogen-bonding interactions with the B helices of HEAT repeats 7-11. N-terminal truncations that remove one or both of the highly conserved Arg13 and Lys18 residues in this moiety severely abrogate binding to Imp β [105,106]. The Arg residues at the tip of Rev helix α_2 in our structural model overlap closely with the N-terminal IBB moiety (**Fig S17C**), accounting for the ability of the IBB domain to compete with Rev for binding to Imp β [43] and raising the possibility that key basic residues in these two molecular cargos mediate similar interactions with Imp β .

Figure S17. Comparison of Imp β /Rev structural model with RanGTP- and cargo-bound complexes of Imp β . (A) Rev bound to the Imp β C-site is predicted to overlap sterically with RanGTP. (B) Comparison with structures of cargo-bound Imp β complexes. (C) The Rev ARM motif is predicted to localize to the same volume as the N-terminal moiety of the Imp α IBB domain.

MINOR SUGGESTIONS:

1. In the ITC experiments, the K_d values should be determined from triplicate experiments. Instead of showing a single data point, ITC titration curves and binding isotherms of three experiments should overlay and show in the figures.

We have performed the additional experimental replicates requested and updated the K_d values accordingly. We have added the ensemble of titration curves and binding isotherms to the revised manuscript as Fig S4. For clarity, we prefer to display the data side-by-side rather to overlay them.

Figure S4. ITC profiles of the binding of Imp β to (A) Rev^{OD} and (B) Rev^{OD} Δ . Three independent replicates are shown for each assay. *Top*, Differential power time course of raw injection heats for a titration of Imp β into the Rev protein solutions. *Bottom*, Normalized binding isotherms corrected for the heat of dilution of Imp β into buffer. The solid line represents a nonlinear least squares fit using a model consisting of two non-symmetric classes of binding sites. Profiles labelled "Exp1" are identical to the panels shown in Fig 4A and B.

2. The docking model proposed is simulated based on the BS3 crosslinking and mutagenesis experiments. It would be great to mutate more arginine to lysine inside the ARM motif and carry out XL-MS using BS3 or DSSO crosslinkers. The results should provide more spatial insights into the binding between Rev and Importin- β .

While the BS3 crosslinks identified by XL-MS allowed us to confirm the existence of two Rev binding sites on Imp β and to localize these within the very approximate N- and C-terminal regions shown in Fig 7F, it became clear from both the HADDOCK and the exhaustive rigid-body docking analyses that the inclusion of these

crosslinks provided relatively little information regarding the precise position and orientation of Rev. This contrasts with our compensatory mutagenesis data, which provided more powerful constraints on the Rev binding configuration. This reflects the fact that the BS3 crosslinking distance constraint used (30 Å) is quite large compared to the dimensions of the Rev helical hairpin domain (approx. 45 Å x 15 Å x 15 Å), whereas the distance constraints derived from the compensatory mutations were much shorter (5-6 Å) (e.g., See **Table S2A**). Indeed, using our model of the Impβ/Rev complex shown in **Fig 11E**, we verified that the hypothetical inclusion of additional crosslinks involving the ARM motif located at the top of the hairpin only modestly reduced the uncertainty in the roll, yaw and pitch angles of Rev within the complex. Thus, we believe that the potential amount of information to be gained from the proposed experiment would not greatly improve our current structural model of the Impβ/Rev complex and hence is not worth the required effort and expense.

3. To simulate the protein interacting model, experiments that provide spatial and temporal information will predict the model more precisely. Hydrogen-Deuterium exchange-MS might give a new insight into the protein dynamics and aid the docking model. In principle, the binding surface between the Rev and importin-β would less dynamic and show relatively low deuterium uptake and the second binding patch for Rev might be revealed on importin-β if existed.

We previously explored HDX-MS experiments during early stages of this project, but preliminary analyses discouraged us from further pursuing these. Briefly, our reason for abandoning this approach was that, since Impβ is a highly flexible molecule that becomes significantly more rigid upon binding cargo, it was not possible to distinguish Impβ residues whose accessibility became altered due to rigidification of the protein from those that directly mediated Rev recognition. Hence, we feel that pursuing HDX-MS analyses would not result in a significant improvement of our current structural model.

4. A schematic model is required to explain how monomeric Rev interacts with Importin-β and is delivered to the nucleus.

We have added the requested scheme to the revised manuscript as **Fig 1A**:

Figure 1. Rev structure and dimerization interfaces. (A) Scheme depicting the nuclear import cycle of Rev by Impβ. **(B)** Domain organization of Rev. **(C)** Oligomerization interfaces reported for Rev. Residues mutated to destabilize the A-A and B-B interfaces are indicated in cyan.

Reviewer #2 (Significance (Required)):

Previous studies have revealed that Rev protein contains nuclear localization signal and is transported to the nucleus to carry out its nucleus functions. Importin- β is one of the nuclear import factors that mediate the nuclear import of Rev in T lymphocytes. Hence, the authors here focus on the interaction of Importin- β and Rev. I appreciate the idea of understanding the molecular details of Importin- β and Rev, the potential for the development of therapeutic chemical inhibitors. However, the atypical binding stoichiometry between Importin- β and Rev authors claimed is not convincing based on their current data. Hence, a substantial revision using materials they have in hand and methods they have established will be required. Importantly, more biological insights will be needed.

Our revised manuscript now includes the additional requested evidence showing that Imp β binds WT Rev with 1:2 stoichiometry. Regarding the question of physiological relevance, our analyses performed in response to point 2c demonstrate that the binding of Rev to the second site on Imp β is significant at Rev concentrations likely to prevail during viral infection. We believe that these revisions have adequately addressed the concerns expressed by Reviewer 2, resulting in a significantly enhanced manuscript.

July 19, 2022

RE: Life Science Alliance Manuscript #LSA-2022-01431R

Dr. Carlo Petosa
Institut de Biologie Structurale
71 Avenue des Martyrs
Grenoble 38000
France

Dear Dr. Petosa,

Thank you for submitting your revised manuscript entitled "Binding stoichiometry and structural model of the HIV-1 Rev/Importin beta complex". We would be happy to publish your paper in Life Science Alliance pending final revisions necessary to meet our formatting guidelines.

- please address Reviewer 1's remaining comments
- please upload your supplementary figures as single files and upload your table files in either editable doc or excel file format
- please add the author contributions to the main manuscript text
- please add your supplementary figure legends and table legends to the main manuscript text
- we encourage you to introduce the panels in your figure legends in alphabetical order
- please add a callout for Figure S1 B to the main manuscript text

Figure Check:

-there appears to be a splice in Figure S1A, after the 2nd lane of the native gel. If this is accurate, please indicate the splice with a vertical line and indicate this in the figure legend.

A. FINAL FILES:

B. MANUSCRIPT ORGANIZATION AND FORMATTING:

Sincerely,

Reviewer #1 (Comments to the Authors (Required)):

I am satisfied by the way the authors took suggestions seriously and carried out additional experiments which greatly expanded the depth of this study. I think that the revised paper is interesting and will be appreciated in the field as it reveals a non-canonical binding between Rev-Importin beta based on a series of in vitro experimental results, and, importantly, suggests how this interaction may occur in virus-infected cells.

I only have two minor suggestions:

1. Authors pointed out that proteins carrying two substrate-binding sites do not necessarily display a biphasic isotherm upon substrate binding, and provided references in the point-to-point response letter. I would suggest also including this information (a few sentences/references) in the ITC section in the main text (Page 6).
2. Importin beta alone crystal structures have been determined and deposited in the protein data bank (e.g. PDB code: 4XRK). It would be great to include predicted SAXS scattering profiles of Importin beta alone in addition to cargo-bound Importin betas as a control.

Reviewer #2 (Comments to the Authors (Required)):

The authors have addressed my minor corrections in the revised manuscript and have made significant effort to address the second reviewer's substantial comments. I consider this a very detailed study, that is very data rich, with solid conclusions. I have no issues regarding suitability for publication.

Point-by-point response

Thank you for submitting your revised manuscript entitled "Binding stoichiometry and structural model of the HIV-1 Rev/Importin beta complex". We would be happy to publish your paper in Life Science Alliance pending final revisions necessary to meet our formatting guidelines.

-please address Reviewer 1's remaining comments

See below.

-please upload your supplementary figures as single files and upload your table files in either editable doc or excel file format

Done.

-please add the author contributions to the main manuscript text

Done.

-please add your supplementary figure legends and table legends to the main manuscript text

Done.

-we encourage you to introduce the panels in your figure legends in alphabetical order

This comment refers to Figures 3 and 5.

We reorganized the panels in Figure 3 so that they are now cited in alphabetical order in the text (old panel G becomes new panel A and old panels A-F become new panels B-G). Regarding Figure 5, the text cites panels in the order A, C, B and D. We have decided to maintain this order because relabelling or reorganizing the figure panels would be awkward and modifying the text to cite panels in alphabetical order would make the logical flow less natural.

In addition, we have renumbered the supplementary Tables so that these are now introduced in numerical order. (Old Table S5 becomes new Table S1, and old Tables S1-S4 become new Tables S2-S5).

-please add a callout for Figure S1 B to the main manuscript text

This is now included in the following sentence:

"A glutaraldehyde crosslinking experiment followed by mass spectrometry (MS) analysis corroborated the ability of Imp β to bind two Rev^{OD} monomers (**Fig S1A and B**)."

Figure Check:

-there appears to be a splice in Figure S1A, after the 2nd lane of the native gel. If this is accurate, please indicate the splice with a vertical line and indicate this in the figure legend.

We have added the vertical line and added the following sentence to the figure legend.

"The vertical black line on the native gel indicates that lanes 1-2 and lanes 3-5 are taken from non-adjacent regions of the same gel."

Reviewer #1 (Comments to the Authors (Required)):

I am satisfied by the way the authors took suggestions seriously and carried out additional experiments which greatly expanded the depth of this study. I think that the revised paper is interesting and will be appreciated in the field as it reveals a non-canonical binding between Rev-Importin beta based on a series of in vitro experimental results, and, importantly, suggests how this interaction may occur in virus-infected cells.

I only have two minor suggestions:

1. Authors pointed out that proteins carrying two substrate-binding sites do not necessarily display a biphasic isotherm upon substrate binding, and provided references in the point-to-point response letter. I would suggest also including this information (a few sentences/references) in the ITC section in the main text (Page 6).

We are pleased that Reviewer 1 is satisfied with the revisions. As requested we have added the following sentence to the ITC section and included two additional references:

"We note that the uniphasic, rather than biphasic, appearance of the observed binding isotherms is compatible with the presence of two binding sites, as has been previously shown [92,93]."

- 92 Freire E, Schon A & Velazquez-Campoy A (2009) Isothermal titration calorimetry: general formalism using binding polynomials. *Methods Enzymol* 455: 127-155. doi:[10.1016/S0076-6879\(08\)04205-5](https://doi.org/10.1016/S0076-6879(08)04205-5)
- 93 Brautigam CA (2015) Fitting two- and three-site binding models to isothermal titration calorimetric data. *Methods* 76: 124-136. doi:[10.1016/j.ymeth.2014.11.018](https://doi.org/10.1016/j.ymeth.2014.11.018)

2. Importin beta alone crystal structures have been determined and deposited in the protein data bank (e.g. PDB code: 4XRK). It would be great to include predicted SAXS scattering profiles of Importin beta alone in addition to cargo-bound Importin betas as a control.

This comment refers to **Fig 6A** and **C**, which compare SAXS scattering profiles and model-independent parameters measured for the human Imp β /HIV-1 Rev^{OD} Δ complex with corresponding theoretical profiles and parameters calculated for eight different Imp β conformations derived from crystal structures of cargo-bound complexes (minus the cargo atoms). Seven of the structures used for these calculations are of human Imp β and the eighth is of murine Imp β , which shares >98% sequence identity with human Imp β . Consequently, differences among the calculated SAXS profiles and parameters reflect changes in the protein conformation. The results show that the agreement between the observed and calculated values is better for extended conformations than for compact conformations.

We agree with the Reviewer that it would be interesting to include an additional conformation of Imp β in the unbound state. However, no such structure is available for human Imp β or for a close ortholog. The 4XRK crystal structure mentioned is that of Imp β from the fungus *Chaetomium thermophilum*, which shares only 38% sequence identity with human Imp β and is 1 kDa lower in mass (96.2 kDa vs 97.2 kDa). Moreover, the first 37 and last 8 residues are missing from the crystal structure, reducing the calculated mass even further (to 91.3 kDa). To our knowledge the only other structure available of unbound Imp β is that of *S. cerevisiae* (PDB 3ND2), which shares only 34% sequence identity with human Imp β and is 2.4 kDa lower in mass (94.8 kDa). Hence, differences observed between SAXS data calculated from these fungal structures and those determined experimentally or calculated from the eight mammalian structures would not only reflect differences in conformation but also in overall mass and sequence. Moreover, the two fungal conformations are similar to conformations already presented in **Fig 6** (3ND2 is comparable to 1QGK, while 4XRK is comparable to 1UKL). We therefore prefer not to include these structures in **Fig 6**.

Reviewer #2 (Comments to the Authors (Required)):

The authors have addressed my minor corrections in the revised manuscript and have made significant effort to address the second reviewer's substantial comments. I consider this a very detailed study, that is very data rich, with solid conclusions. I have no issues regarding suitability for publication.

We thank Reviewer #2 for having evaluated our manuscript.

July 22, 2022

RE: Life Science Alliance Manuscript #LSA-2022-01431RR

Dr. Carlo Petosa
Institut de Biologie Structurale
71 Avenue des Martyrs
Grenoble 38000
France

Dear Dr. Petosa,

Thank you for submitting your Research Article entitled "Binding stoichiometry and structural model of the HIV-1 Rev/Importin beta complex". It is a pleasure to let you know that your manuscript is now accepted for publication in Life Science Alliance. Congratulations on this interesting work.

DISTRIBUTION OF MATERIALS:

Again, congratulations on a very nice paper. I hope you found the review process to be constructive and are pleased with how the manuscript was handled editorially. We look forward to future exciting submissions from your lab.

Sincerely,
